# Freezing stress affects the efficacy of clodinafop-propargyl and 2,4-D plus MCPA on wild oat (*Avena ludoviciana* Durieu) and turnipweed [*Rapistrum rugosum* (L.) All.] in wheat (*Triticum aestivum* L.)

**Alireza Hasanfard[1], Mehdi Rastgoo[1]\*, Ebrahim Izadi Darbandi[1], Ahmad Nezami[1], Bhagirath Singh Chauhan[2]**

**1** Department of Agrotechnology, Faculty of Agriculture, Ferdowsi University of Mashhad, Mashhad, Iran,
**2** Queensland Alliance for Agriculture and Food Innovation (QAAFI) and School of Agriculture and Food Sciences (SAFS), The University of Queensland, Gatton, Queensland, Australia

\* m.rastgoo@um.ac.ir

**Data Availability Statement:** All relevant data are within the article.

## Abstract

The occurrence of freezing stress around herbicides application is one of the most important factors influencing their performance. This experiment was performed to evaluate the efficacy of clodinafop-propargyl and 2,4-D plus MCPA (2,4-Dichlorophenoxyacetic acid plus 2-methyl-4-chlorophenoxyacetic acid), the most important herbicides used in wheat fields in Iran, under the influence of a freezing treatment (-4˚C). Wheat and its two common weeds, winter wild oat (*Avena ludoviciana* Durieu) and turnipweed [*Rapistrum rugosum* (L.) All.], were exposed to the freezing treatment for three nights from 7:00 P.M. to 5:00 A.M. before and after herbicide application, and their response was compared with plants that did not grow under freezing stress. Under no freezing (NF) and freezing after spray (FAS) conditions, winter wild oat was completely controlled with the recommended dose of clodinafop-propargyl (64 g ai ha$^{-1}$; hereafter g ha$^{-1}$). However, the survival percentage of winter wild oat in the freezing before spray (FBS) of clodinafop-propargyl 64 g ha$^{-1}$ was 7%, and it was completely controlled with twice the recommended dose (128 g ha$^{-1}$). Under NF conditions and FAS treatment, turnipweed was completely controlled with twice the recommended dose of 2,4-D plus MCPA (2025 g ae ha$^{-1}$; hereafter g ha$^{-1}$), while there was no complete control under recommended rate. However, in the FBS treatment, the survival of turnipweed was 7% under double dose. The LD$_{50}$ (dose required to control 50% of individuals in the population) and GR$_{50}$ (dose causing 50% growth reduction of plants) rankings were NF<FBS<FAS for clodinafop-propargyl and NF<FAS<FBS for 2,4-D plus MCPA. Selectivity index for clodinafop-propargyl in NF conditions, FBS, and FAS treatments was 2.4, 0.91, and 0.78, respectively, and, for 2,4-D plus MCPA, it was 2.6, 0.12, and 0.88, respectively. According to the results of LD$_{50}$, it can be stated that the freezing stress after the spraying of clodinafop-propargyl and before the spraying of 2,4-D plus MCPA would further reduce the efficacy of these herbicides.

**Funding:** We are grateful to Ferdowsi University of Mashhad, Iran, for providing financial support for present research (Project No.3.49759). No additional external funding was received for this study. The funders (Ferdowsi University of Mashhad, Iran and University of Queensland) had no role in study design, data collection and analysis, decision to publish, or preparation of the manuscript.

**Competing interests:** The authors have declared that no competing interests exist.

## Introduction

Winter wild oat (*Avena ludoviciana* Durieu) and turnipweed [*Rapistrum rugosum* (L.) All.] are the most important annual weeds of winter wheat (*Triticum aestivum* L.) in Iran, which cause significant reductions in wheat yield [1, 2]. The high reproduction potential and fitness to freezing stress, which results in a wide dispersal of weed seeds are the major challenges in the management of winter annual weeds like wild oat and turnipweed. Therefore, in cold regions of Iran, the dispersal and geographical distribution of these weeds are more significant than other weed species [3].

One of the most important consequences of climate change is the change in glaciers' occurrence patterns [4]. A sharp drop in temperature after high temperatures leads to crop damage. Increasing autumn temperatures in Iran, resulting from climate change, have led to reduced freezing tolerance of autumn crops [3]. In other words, autumn crops will not tolerate freezing stress due to the lack of hardening process in autumn. As a result, they will be damaged by decreasing temperatures during winter, and their yield will be reduced [3, 4]. It has been reported that increasing autumn temperatures resulting from climate change in western Canada led to reduced freezing tolerance of winter cereals [5]. In other words, due to the lack of optimal cold acclimation in autumn, crops will not be able to tolerate freezing stress. As a result, they will be damaged by decreasing temperatures during winter or by the occurrence of late spring frosts [6]. However, weeds generally have a higher ability to tolerate freezing stress [7]. Based on this fact, it can be inferred that climate change is adversely changing agricultural production and can significantly change the management of current cropping systems that have evolved under optimum climatic conditions.

Hand weeding and hoeing are not common in wheat fields, and mechanical weed management methods are not effective, so chemical methods are mainly used to control weeds in the wheat fields in Iran. Clodinafop-propargyl and 2,4-D plus MCPA (2,4-Dichlorophenoxyacetic acid plus 2-methyl-4-chlorophenoxyacetic acid) are the most common herbicides used for controlling weeds in wheat fields in Iran. Due to the particular importance of this crop in food security, their wide excessive applications can be justified [8, 9]. Clodinafop-propargyl is a systemic and selective herbicide of the aryloxyphenoxypropionate (APP) group and used post-emergent to control annual grass weeds in wheat. This herbicide inhibits the activity of acetyl-coenzyme A carboxylase (ACCase) and thus disrupts the biosynthesis of fatty acids [10]. 2,4-D plus MCPA, synthetic auxin herbicides, is a systemic herbicide mixture registered for selective post-emergence control of annual broadleaf weeds in wheat [11].

The efficacy of herbicides is significantly affected by environmental conditions before, during, and after application [12]. It has been reported that environmental factors such as light intensity, temperature, water stress, relative humidity, and nutrient availability alter the herbicide efficacy [13, 14]. On the other hand, plant species and their growth stages also affect herbicide efficacy [15]. Numerous studies indicate that ambient temperature is one of the most influential factors affecting the performance of herbicides [14, 16, 17]. Increasing the temperature is likely to increase the absorption and translocation of some herbicides and increase their performance. In contrast, the improvement in herbicide performance at low temperatures may be due to a reduction in the rate of herbicide metabolism [17, 18]. In cold regions, freezing temperatures occur around the time of herbicide application. Therefore, the occurrence of severe freezing that causes damage to plant leaves will have adverse effects on the activity and efficacy of foliar-applied herbicides [19]. On the other hand, application of herbicides in unfavorable environmental conditions (such as adverse temperature before, during, and after herbicide application) causes stress in the crop, and this changes the selectivity index of the herbicide, which will ultimately cause damage to the crop [20]. The selectivity index refers to

the potential of a herbicide to eliminate weeds in a crop without affecting the yield or quality of the final product [21].

At the appropriate growth stage of weeds for herbicide application, the temperature usually decreases, and this issue leads to a decrease in the performance of herbicides. Due to the lack of information on the efficacy of herbicides under freezing stress, this experiment was conducted to evaluate the efficacy of clodinafop-propargyl and 2,4-D plus MCPA on wild oat and turnipweed under freezing stress before and after herbicide application.

## Materials and methods

### Experimental site and plant material

Seeds of wheat (cv. Mihan) were harvested from Khorasan Razavi Agricultural and Natural Resources Research and Education Center, and seeds of winter wild oat and turnipweed were collected (collected from about 500 plants) from infested farms around Mashhad-Iran in May 2019. Experiments were conducted at the Faculty of Agriculture, the Ferdowsi University of Mashhad, Iran (Lat 36° 15′ N, Long 59° 28 E; 985 m Altitude). To break the dormancy of winter wild oat and turnipweed, their seeds were exposed to moist-chilling treatment and potassium nitrate ($KNO_3$) 0.2% solution for 3 days at 5°C in the dark, respectively [3]. Weed seeds were placed in 9-cm diameter Petri dishes (20–25 seeds per Petri dish) on moist filter papers and kept at 20°C. After 5 days, the germination percentage of winter wild oat and turnipweed seeds was >90%. The experiment was repeated twice with three replicates. To experience the same cold acclimation, the plants were planted at approximately the same time on October 21 (first run) and 24 (second run) in 2019. Plastic pots (12 cm in diameter and 20 cm in height) were used in the trial, and pots were filled with a mixture of sand, agricultural soil, and peat moss (2:2:1 sand, agricultural soil, and peat moss, respectively). For uniform emergence, turnipweed germinated seeds were planted in a seedling tray. Eight turnipweed seedlings were transferred to pots in the cotyledon stage. Eight germinated seeds of wild oat were planted at 4–5 cm depth below the soil surface. Wheat seeds were planted directly in each pot at the same depth. To induce cold acclimation, the plants were kept outdoors, grown in natural conditions (Fig 1), and irrigated according to daily requirements until they were tested.

### Experimental factors

Experimental factors included freezing stress at three conditions, including no freezing (NF), freezing before spray (FBS), and freezing after spray (FAS). Plant species included wheat, winter wild oat, and turnipweed. Herbicides were applied at 0, 6.25, 12.5, 25, 50, 100, and 200% of the recommended dose for the herbicide clodinafop-propargyl (0.8 L ha⁻¹; Topik, 8% EC) and 2,4-D plus MCPA (1.5 L ha⁻¹; U 46 combi fluid, 67.5% SL).

### Freezing treatment and herbicide application

Under NF, the plants grew completely in natural winter conditions without exposing to freezing stress. For FBS, the plants grew up to the 3–4 leaf stage under natural conditions. They were exposed to freezing from 7:00 P.M. to 5:00 A.M. for three nights before herbicide application [19]. Freezing treatment was performed using a thermogradient freezer (Weiss Technik). The initial temperature of the freezer was 5°C; however, the freezer was cooled down at a rate of 2°C per hour to reach a temperature of -4°C. The temperature drop slope was adjusted based on the long-term meteorological statistics of Mashhad in the time range of herbicide application. The pots were kept at -4°C for 1 h and then removed from the freezer (Fig 2).

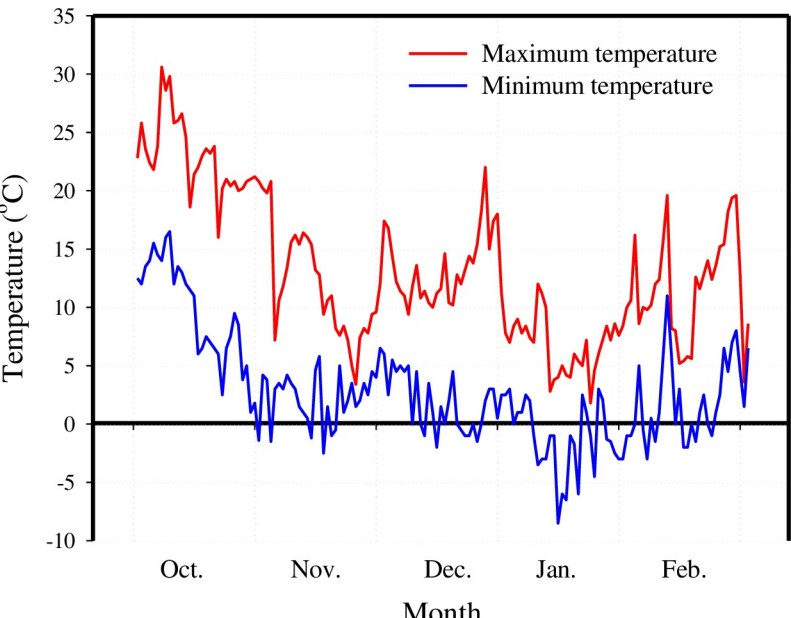

**Fig 1. Minimum and maximum daily air temperatures at the experimental site from October 2019 to February 2020.** Relative humidity of 40–50% under the natural photoperiod [3].

Each morning at 5:30 A.M., the plants were transferred out of the freezer, kept in natural conditions until 6:30 P.M., and then returned to the freezer during the night. After three nights of freezing exposure, the pots were taken out of the freezer and sprayed with herbicides after 2 h [19]. After spraying, the plants were kept outside under natural conditions for the duration of the experiment.

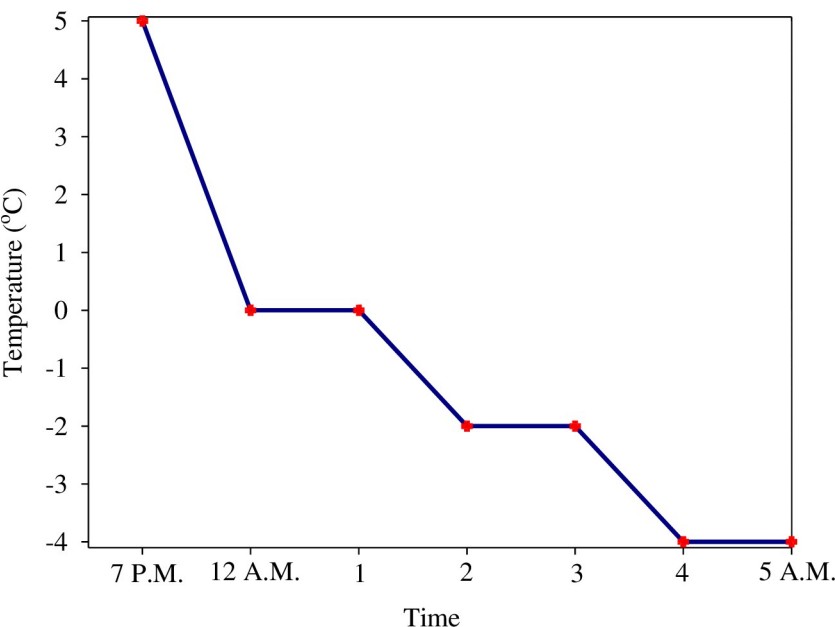

**Fig 2. Trend of temperature decrease in thermogradient freezer during the night from 7:00 P.M. to 5:00 A.M.**

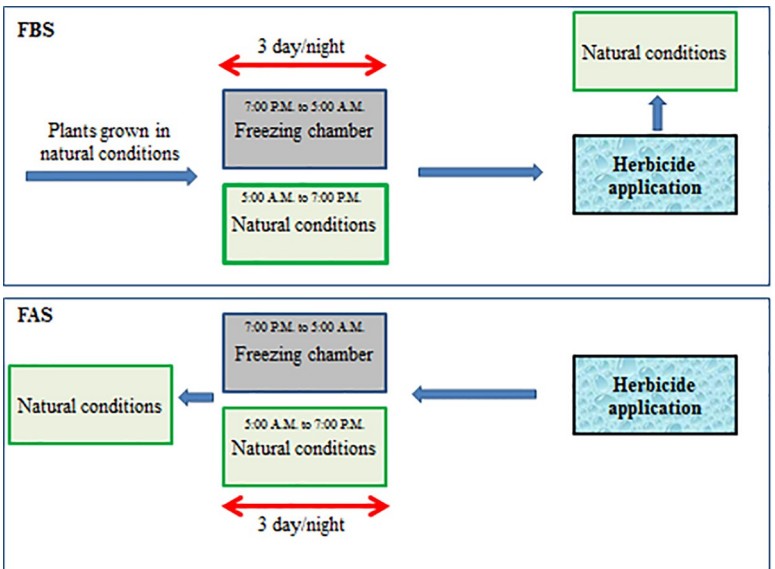

**Fig 3. Schematic diagram of the experimental study design.** Abbreviation: FBS (freezing before spray); FAS (freezing after spray).

In the FAS treatment, freezing stress was applied for three nights by placing the samples treated with herbicides in the thermogradient freezer. For this purpose, the plants were taken out of the freezer at 5:30 A.M., transferred to natural conditions, and then returned to the freezer at 6:30 P.M. for the overnight freezing treatment (Similar to FBS treatment). Three nights after freezing treatment, the plants were transferred to natural conditions to continue the trial process (Fig 3).

The conditions inside the thermogradient freezer were the same for both freezing treatments before and after spray. The experiment was performed in the freezer in completely dark conditions and at the same relative humidity (35±5%) for all treatments. Temperature changes in the freezer were regularly monitored when the samples were in the freezer. For all the freezing treatments, herbicides were applied at the same time. Plants were transferred to natural conditions after three nights of freezing treatment at 5:30 A.M. Also, plants related to freezing after spray were transferred to the freezer at 6:30 P.M. On the same day, plants that were not exposed to freezing were sprayed from 7:30 A.M. to 10.30 A.M. The herbicides were applied by a researcher-made moving boom sprayer equipped with a TeeJet 8002 flat fan nozzle (spray output volume of 200 L ha$^{-1}$ at a pressure of 260 kPa). Plants were returned and maintained outdoor under natural conditions of winter after the treatment.

## Measurement of traits

Four weeks after herbicide treatments, survival percentage (Eq 1) and shoot dry weight were measured by placing the samples in an oven at 75°C for 48 h. Plants that produced new leaves were considered alive and those that experienced severe chlorosis and stunted growth were recorded as dead [22].

$$Survival \% = \frac{N1}{N0} \times 100 \tag{Eq1}$$

Where, $N_1$ and $N_0$ are the numbers of alive plants four weeks after herbicide treatment and the number of alive plants before herbicide treatment, respectively.

## Dose–response experiments

The response of weed traits to the doses of herbicides tested was analyzed by nonlinear regression. For this purpose, the data were fitted to a three-parameter log-logistic equation (Eq 2) [23].

$$Y = \frac{d}{1 + e^{b(\log\ x - loge)}} \qquad \text{(Eq2)}$$

Where, Y is the response rate (survival % or dry weight) at dose x, d is the upper limit for Y, b is the slope of the curve around e, and e is the effective dose (g ai or ae ha$^{-1}$). Here, e is replaceable with $LD_{50}$ (dose required to control 50% of individuals in the population) or $GR_{50}$ (dose causing 50% dry weight reduction of plants).

## Relative potency

Relative potency was used to determine herbicide efficacy under freezing stress and no freezing stress (Eq 3) [24].

$$Relative\ potency = \frac{ED_{50A}}{ED_{50B}} \qquad \text{(Eq3)}$$

Where, $ED_{50A}$ is the dose required to reduce survival by 50% in NF, and $ED_{50B}$ is the dose needed to reduce survival by 50% in FBS or FAS. Relative potency = 1 indicates that freezing treatment (FBS and FAS) did not affect herbicide efficacy. If relative potency > 1, it indicates that the herbicides are more efficient under freezing stress. Relative potency < 1 indicates the decreased effectiveness of the herbicides under freezing stress with respect to optimum conditions.

## Selectivity index

The ratio between doses that caused 10% damage to crops and 90% damage to weeds was considered a selectivity index [24] and calculated using Eq 4.

$$Selectivity\ index\ (10, 90) = \frac{ED_{10\ (crop)}}{ED_{90\ (weed)}} \qquad \text{(Eq4)}$$

Where, $ED_{90}$ is 90% of the herbicide effect on weed survival (*herbicide* dose to kill 90% of the population), and $ED_{10}$ is 10% of the herbicide impact on crop survival (a 10% effect on the crop). The higher the SI, the more selective the herbicide in the crop [24].

## Statistical analysis

The experiments were conducted using a completely randomized design in factorial arrangement with three replications. Afterward, differences between treatments were compared with the standard error of mean (SEM) values. For data analysis, analysis of variance (ANOVA) was performed using the SAS software (v. 9.4, SAS Institute Inc, Cary, NC, USA). Data for both runs were pooled as no significant differences were observed between runs (P > 0.05). Homogeneity of variance was confirmed using Levene's test. Analysis of biometric data was performed by the R program (v. 4.0.4) [25]. Dose-response curves were also fitted with the asymmetrical sigmoid form. $LD_{50}$ and $GR_{50}$ were separated using 95% confidence intervals (CI).

## Results

### Clodinafop-propargyl

**Survival percentage and $LD_{50}$.** The dose-response experiments were similar for NF and FAS treatments at the recommended dose (Fig 4A). Thus, in both conditions, winter wild oat was completely controlled with the recommended dose (64 g ai $ha^{-1}$). However, the survival of seedlings in the FBS treatment was 7%, and to achieve complete control, 2-fold the recommended dose (128 g $ha^{-1}$) was needed. The $LD_{50}$ in NF, FBS, and FAS was 17.9, 18.1, and 23.2 g $ha^{-1}$, respectively (Table 1).

$LD_{50}$ and $GR_{50}$ were separated using their 95% confidence intervals (CI). The data shows the parameter's mean ± SE. The data ($LD_{50}$ and $GR_{50}$) with the different letters indicate significant differences according to LSD at P ≤0.05. n = 6.

**Dry weight and $GR_{50}$.** The dry weight of winter wild oat in NF and FAS with the recommended dose (64 g $ha^{-1}$) and FBS with twice the recommended dose (128 g $ha^{-1}$) reached 0 g (Fig 4B). The $GR_{50}$ of plants not exposed to freezing stress was 10.6 g $ha^{-1}$. The $GR_{50}$ was 11.6 and 12.8 g $ha^{-1}$ in FBS and FAS, respectively (Table 1).

### 2,4-D plus MCPA

**Survival percentage and $LD_{50}$.** The dose-response experiments of 2,4-D plus MCPA (similar to clodinafop-propargyl) were the same in NF and FAS treatments (Fig 5A). In both conditions, turnipweed was completely controlled with twice the recommended dose (2025 g ae $ha^{-1}$). However, in the FBS treatment, the survival of turnipweed was 7%. At the recommended dose (1012.5 g $ha^{-1}$), the survival of turnipweed in NF, FBS, and FAS was 3%, 33%, and 20%, respectively. The $LD_{50}$ in NF, FBS, and FAS was 293, 473, and 319 g $ha^{-1}$, respectively (Table 1).

**Dry weight and $GR_{50}$.** The dry weight of turnipweed in NF and FAS with twice the recommended dose (2025 g $ha^{-1}$) of 2,4-D plus MCPA was reduced to 0 (Fig 5B). However, in the FBS treatment, the dry weight of turnipweed with twice the recommended dose was 0.012 g per pot. At the recommended dose, the dry weight of turnipweed in FBS was 97% and 76% more than NF and FAS, respectively. The $GR_{50}$ of NF, FBS, and FAS were 176, 299, and 208 g $ha^{-1}$, respectively (Table 1).

### Relative potency

The relative potency of clodinafop-propargyl and 2,4-D plus MCPA was significantly reduced in FBS and FAS compared to NF (Fig 6). Thus, the relative potency in FBS and FAS of clodinafop-propargyl was 0.99 and 0.77, respectively, and at 2,4-D plus MCPA was 0.62 and 0.92, respectively.

### Selectivity index

As shown in Table 2, selectivity index values for clodinafop-propargyl in NF, FBS, and FAS treatments were 2.4, 0.91, and 0.78, respectively, and, for 2,4-D plus MCPA, they were 2.6, 0.12, and 0.88, respectively. For clodinafop-propargyl, the $LD_{10}$ of wheat in NF was 127.9 g $ha^{-1}$. However, $LD_{10}$ of wheat in the FBS and FAS was 53.5 and 57.1 g $ha^{-1}$, respectively (Table 2). For 2,4-D plus MCPA, the $LD_{10}$ of wheat in NF, FBS, and FAS was 2380, 264, and 1236 g $ha^{-1}$, respectively.

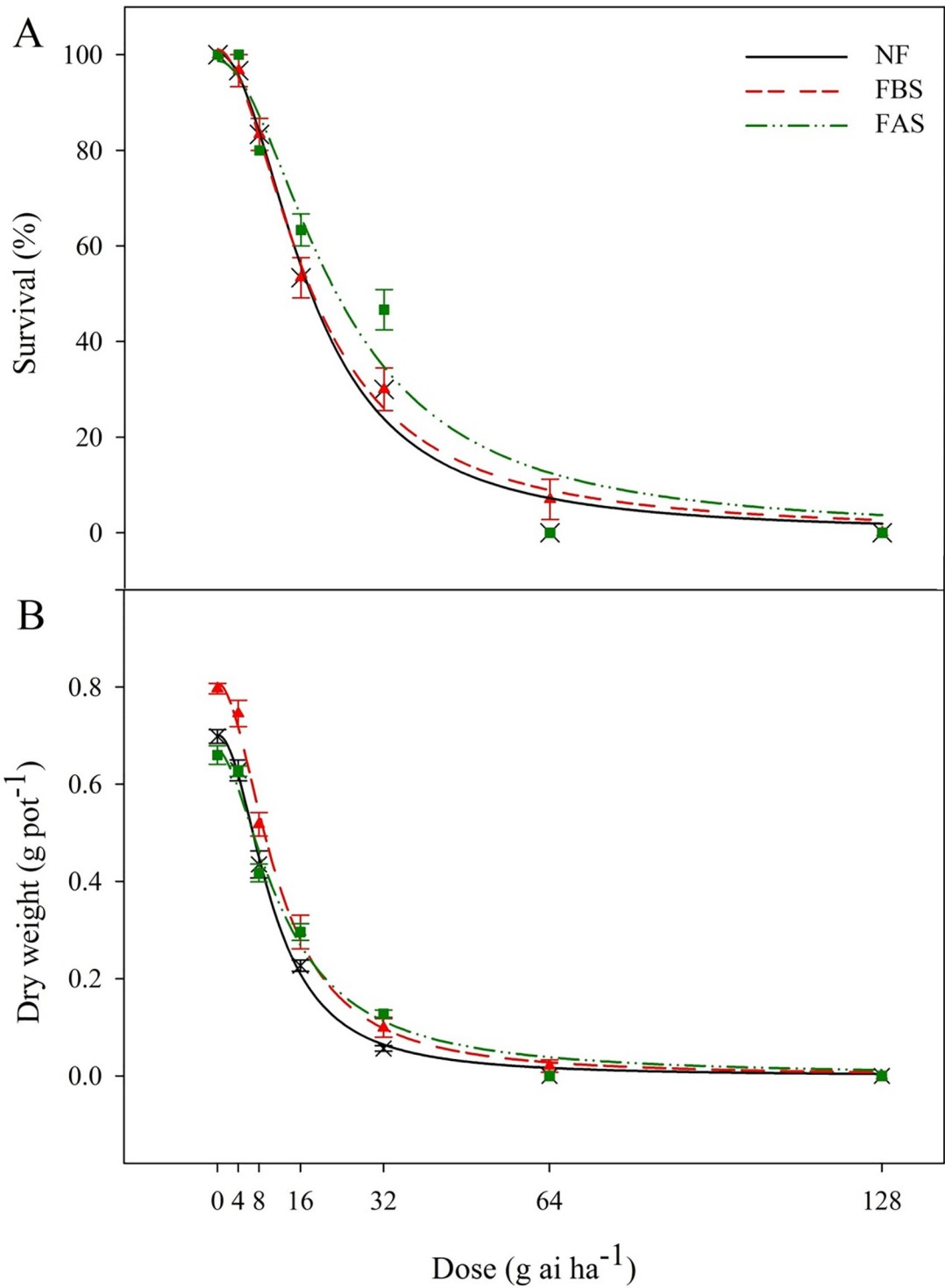

**Fig 4.** Survival (A) and dry weight (B) of winter wild oat treated with various doses of clodinafop-propargyl under different freezing treatments. Abbreviation: NF (no freezing); FBS (freezing before spray): FAS (freezing after spray). The recommended dose of clodinafop-propargyl in Iran is 64 g ha⁻¹.

**Table 1. The dose of clodinafop-propargyl and 2,4-D plus MCPA required for 50% mortality (LD$_{50}$) and the dose that caused 50% inhibition of growth noted by biomass (GR$_{50}$) of winter wild oat and turnipweed under different freezing treatments.**

| Herbicide | Freezing Treatment | b±SE | d±SE | LD$_{50}$ ±SE | 95% CI | R$^2$ |
|---|---|---|---|---|---|---|
| | | | % | g ai or ae ha$^{-1}$ | | |
| Clodinafop-propargyl | NF | 2.0 ± 0.19** | 100.3 ± 2.8** | 17.9 ± 1.2**b | 15.6–20.2 | 0.99 |
| | FBS | 1.9 ± 0.17** | 101.0 ± 2.9** | 18.1 ± 1.2**b | 15.7–20.5 | 0.99 |
| | FAS | 1.9 ± 0.19** | 98.7 ± 2.9** | 23.2 ± 1.7**a | 19.8–26.6 | 0.96 |
| 2,4-D plus MCPA | NF | 1.9 ± 0.20** | 102.2 ± 3.2** | 292.8 ± 21.9**c | 249.4–336.3 | 0.98 |
| | FBS | 1.5 ± 0.15** | 102.4 ± 3.2** | 472.8 ± 42.4**a | 388.8–556.8 | 0.98 |
| | FAS | 1.5 ± 0.14** | 102.8 ± 3.5** | 318.8 ± 28.9**b | 261.6–376.1 | 0.98 |
| | | b±SE | d±SE | GR$_{50}$ ±SE | 95% CI | |
| | | | g pot$^{-1}$ | g ai or ae ha$^{-1}$ | | |
| Clodinafop-propargyl | NF | 2.1 ± 0.15** | 0.70 ± 0.02** | 10.6 ± 0.52**c | 9.6–11.6 | 0.99 |
| | FBS | 2.0 ± 0.12** | 0.80 ± 0.02** | 11.6 ± 0.51**b | 10.6–12.6 | 0.99 |
| | FAS | 1.7 ± 0.12** | 0.67 ± 0.02** | 12.8 ± 0.74**a | 11.3–14.3 | 0.98 |
| 2,4-D plus MCPA | NF | 2.2 ± 0.23** | 0.46 ± 0.01** | 176.2 ± 11.3**c | 153.9–198.6 | 0.99 |
| | FBS | 1.7 ± 0.16** | 0.48 ± 0.01** | 298.6 ± 21.5**a | 256.0–341.3 | 0.99 |
| | FAS | 1.6 ± 0.15** | 0.47 ± 0.01** | 207.8 ± 15.9**b | 176.2–239.4 | 0.99 |

Abbreviation: NF (no freezing); FBS (freezing before spray); FAS (freezing after spray); SE: Standard Error. b is the slope of the curve around e (LD$_{50}$ and GR$_{50}$); d is the upper limit for Y (survival % or dry weight)

**: significant at p≤0.01.

## Discussion

The required dose of clodinafop-propargyl for complete control of winter wild oat seedlings in FBS treatment was twice that of NF and FAS treatments (Fig 4A). The trend of the reducing survival % in seedlings related to the FBS treatment and NF was similar. The LD$_{50}$ for clodinafop-propargyl in freezing treatments shows that exposure of winter wild oat seedlings to freezing treatment for three nights before and after spray reduces the efficacy of this herbicide compared to NF. The LD$_{50}$ was 22% and 23% higher in the FAS compared to the FBS and NF, respectively (Table 1). Freezing treatment for three nights before and after spraying of clethodim herbicide (the mode of action similar to clodinafop-propargyl) led to an increase in LD$_{50}$ [19]. It decreased herbicide efficacy in rigid ryegrass (*Lolium rigidum* Gaud.) populations compared to NF.

The dose of clodinafop-propargyl required to reduce winter wild oat biomass to zero in the FBS treatment was twice that of the NF and FAS treatments (Fig 4B). The GR$_{50}$ is more pronounced in FBS and FAS compared to NF, indicating a decrease in the efficacy of clodinafop-propargyl. The GR$_{50}$ was 17% and 9% higher in the FAS treatment compared to NF and FBS, respectively. Hence, according to the results of GR$_{50}$, three nights of freezing stress after herbicide application reduced the performance of clodinafop-propargyl more than other treatments (Table 1). It has been reported that wild oat plants (*Avena fatua*) that grew at low temperatures were less sensitive to difenzoquat than plants that grew at high temperatures [26].

Our results clearly show that the occurrence of freezing stress around spray reduces the efficacy of clodinafop-propargyl, especially in the treatment of FAS. Herbicide translocation in the phloem in freezing conditions could decline due to damage to the phloem cells, which may lead to a decrease in the herbicide efficacy. Decreased efficacy of sethoxydim at low temperatures due to reduced herbicide translocation for Quackgrass [*Elymus repens* (L.) Gould] control was reported by Coupland [27]. In general, to maintain the herbicide efficacy under cold

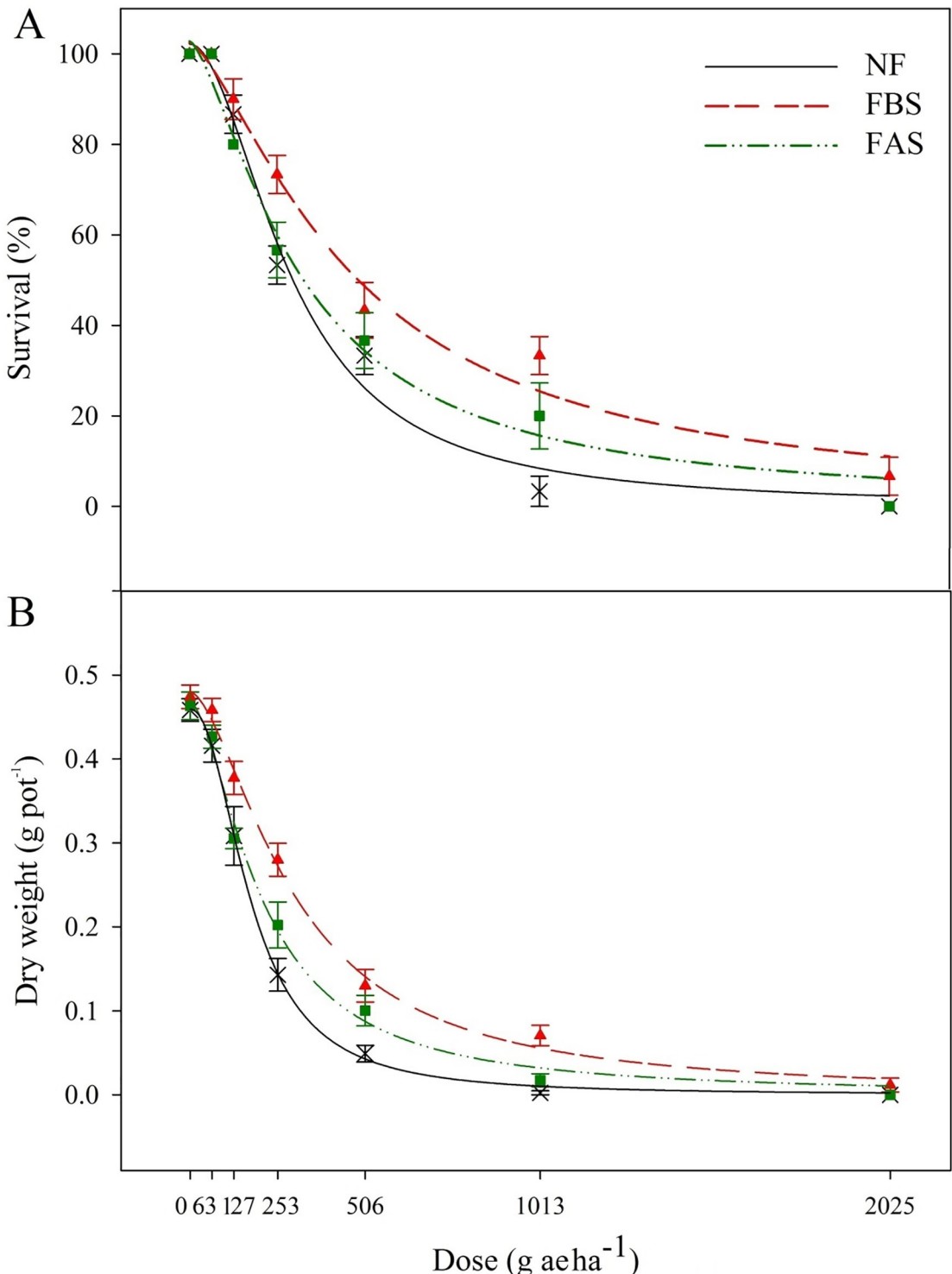

**Fig 5.** Survival (A) and dry weight (B) of turnipweed treated with various doses of 2,4-D plus MCPA under different freezing treatments. Abbreviation: NF (no freezing); FBS (freezing before spray): FAS (freezing after spray). The recommended dose of 2,4-D+MCPA in Iran is 1012.5 g ha⁻¹.

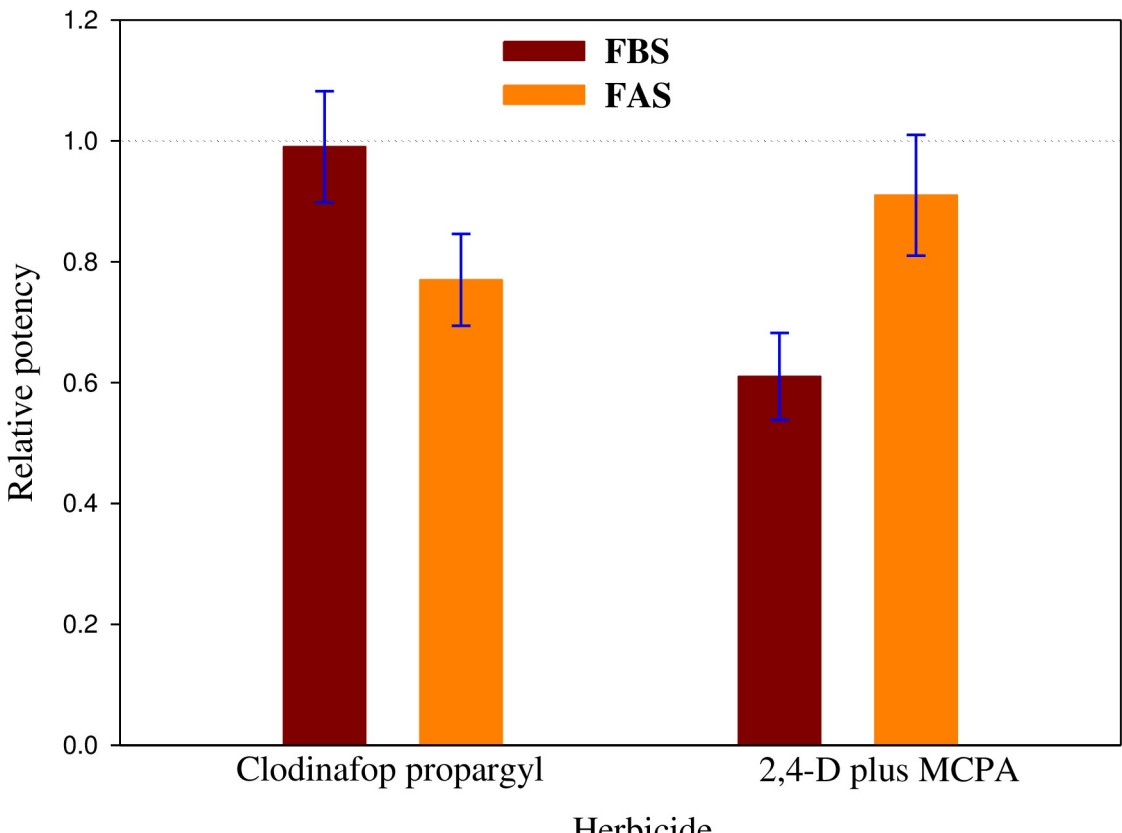

**Fig 6. Relative potency of clodinafop-propargyl and 2,4-D plus MCPA under different freezing treatments.** Abbreviation: FBS (freezing before spray) and FAS (freezing after spray). The error bars indicate standard error (SE). The dashed line represents the relative potency = 1.0.

conditions, using clodinafop-propargyl after the freezing stress can be recommended in areas with a similar climate.

Although the three-night treatment of FBS and FAS resulted in a reduction in turnipweed control by 2,4-D plus MCPA compared to NF, this reduction was different in the two freezing treatments (Fig 5A). At the recommended dose (1013 g ha$^{-1}$) in NF, almost all turnipweed plants were controlled, while the survival percentage of turnipweed in FBS and FAS treatments were 33% and 20%, respectively. Low temperatures around the time of herbicide application

**Table 2. Sensitivity of plant species of wheat, winter wild oat, and turnipweed to clodinafop-propargyl and 2,4-D plus MCPA under different freezing treatments.**

| Freezing Treatment | Plant species | Clodinafop-propargyl | | 2,4-D plus MCPA | |
|---|---|---|---|---|---|
| | | LD$_{10}$ (wheat) and LD$_{90}$ (winter wild oat) (g ai ha$^{-1}$) ± SE | SI | LD$_{10}$ (wheat) and LD$_{90}$ (turnipweed) (g ae ha$^{-1}$) ± SE | SI |
| NF | Wheat | 127.9 ± 8.8 | - | 2380 ± 455 | - |
| | Weed | 53.5 ± 5.4 | 2.4 | 908 ± 108 | 2.6 |
| FBS | Wheat | 53.5 ± 11.0 | - | 264 ± 61.0 | - |
| | Weed | 59.0 ± 6.4 | 0.91 | 2144 ± 327 | 0.12 |
| FAS | Wheat | 57.1 ± 8.7 | - | 1236 ± 163 | - |
| | Weed | 73.6 ± 7.2 | 0.78 | 1400 ± 199 | 0.88 |

Abbreviation: NF; No freezing, FBS: Freezing before spray, FAS: Freezing after spray SI: Selectivity index. The selectivity index (SI) was calculated as follows: *Selectivity index (10, 90):* $\frac{ED10\ (crop)}{ED90\ (weed)}$.

reduced the effectiveness of systemic herbicides [28]. Based on $LD_{50}$ values, the efficacy of 2,4-D plus MCPA under FBS and FAS was 1.6 and 1.1 times lower compared to NF, respectively (Table 1).

FAS is possible to damage the phloem cells, which reduces the translocation of the herbicide. The efficacy of 2,4-D plus MCPA in the treatment of FBS was reduced. The decrease in the effect of 2,4-D plus MCPA in the treatment of FBS is probably due to the reduction of absorption and translocation of this herbicide in turnipweed seedlings. In this regard, an increase in efficacy of 2,4-D and glyphosate at high temperatures compared to low temperatures in common ragweed (*Ambrosia artemisiifolia* L.) and giant ragweed (*Ambrosia trifida* L.) due to the increased absorption and/or translocation at high temperatures around herbicide application has been reported by Ganie et al. [29]. It has been also reported that low temperatures reduced glufosinate activity and translocation in wild radish (*Raphanus raphanistrum* L.) [30].

Turnipweed's dry weight at the recommended dose of 2,4-D plus MCPA (1012.5 g ha$^{-1}$) in FBS and FAS treatments was 35.5 and 8.5 times higher than the NF, respectively (Fig 5B). Also, $GR_{50}$ in FBS and FAS was 41% and 15.2% higher compared to NF, respectively (Table 1). Although a decrease in herbicide efficacy is evident in both FBS and FAS treatments, high $GR_{50}$ in FBS shows an adverse effect of three nights of freezing before the application of 2,4-D plus MCPA. In other words, the efficacy of this herbicide reacts more to FBS than to FAS. In a similar study, the $GR_{50}$ for 2,4-D increased at low temperature in common ragweed compared to high temperature [29].

The freezing stress around 2,4-D plus MCPA application reduced the efficacy of this herbicide in FBS and FAS treatments compared to NF. The decrease in the efficacy of this herbicide in the treatment of FBS is probably due to the reduction of absorption and translocation of the herbicide in the phloem during freezing. Absorption and translocation of 2,4-D plus MCPA might be reduced at low temperatures due to the effect on herbicide penetration facilitated by physicochemical factors, including the decreased rate of diffusion, increased viscosity of the cuticle, and physiological factors comprising decreases in photosynthesis, phloem translocation, and protoplasmic streaming [29]. In Iranian wheat fields, using the 2,4-D plus MCPA before the freezing stress can be recommended to maintain the herbicide efficacy under cold conditions.

When relative potency > 1, the herbicide efficacy is higher than optimal non-freezing conditions, and when relative potency < 1, the herbicide efficacy is lower than usual [24]. The relative potency values of clodinafop-propargyl and 2,4-D plus MCPA were lower than one under both FBS and FAS treatments in the present experiment (Fig 6). In other words, freezing stress reduced the efficacy of these herbicides. Based on the results of relative potency, the reduction of clodinafop-propargyl efficacy in the FAS treatment (33%) was more significant than in the FBS treatment (1%). Conversely, for 2,4-D plus MCPA, the reduction in efficacy was more significant in the FBS treatment (38%) compared to the FAS treatment (8%).

The selectivity index for clodinafop-propargyl and 2,4-D plus MCPA was > 1 in NF, indicating that both herbicides acted selectively in wheat (Table 2). However, applying three nights of FBS and FAS of both herbicides reduced the selectivity index to less than one. In other words, freezing stress reduced the selectivity index of both herbicides, which led to phytotoxicity in wheat. For clodinafop-propargyl, $LD_{10}$ of wheat in NF was 58% and 55% higher than FBS and FAS, respectively. The $LD_{10}$ values of 2,4-D plus MCPA in NF were 89% and 48% higher compared to FBS and FAS, respectively. In other words, the dose required to kill 10% of wheat in both herbicides under NF was higher than FBS and FAS. Therefore, FBS and FAS increased the sensitivity of wheat to herbicide application. Freezing stress may reduce the rate of herbicide metabolism in the crop and ultimately increase its susceptibility [17]. Safeners

decrease herbicide sensitivity in plant species by accelerating the metabolism of herbicides into less active or inactive compounds [31]. The researchers reported that safeners enhance the tolerance of selected grass crops such as wheat to aryloxyphenoxypropionate herbicides [32]. Hence, one of the practical methods to improve the selectivity index of the clodinafop-propargyl during freezing stress is using safeners.

## Conclusions

The relative potency results for both herbicides were used to confirm the results of $LD_{50}$ and $GR_{50}$. Thus, $LD_{50}$ and $GR_{50}$ were higher for clodinafop-propargyl in the FAS treatment than in NF and FBS treatments. The $LD_{50}$ and $GR_{50}$ were higher for the 2,4-D plus MCPA in the FBS treatment than in the NF and FAS treatments. In other words, clodinafop-propargyl was more effective than other conditions in the treatment of three nights of FAS and 2,4-D plus MCPA in the treatment of three nights of FBS. The selectivity index for clodinafop-propargyl was lower in the FAS treatment (0.78) and for 2,4-D plus MCPA was lower in the FBS treatment (0.12). The efficacy of clodinafop-propargyl in the FAS treatment and the efficacy of 2,4-D plus MCPA in the FBS treatment were reduced. The values of the selectivity index of herbicides in the conditions of freezing stress showed that this index could be well used to determine the selective performance of these herbicides. Plant growth regulators and safeners are recommended to increase the selectivity properties of herbicides during exposure to freezing and reduce the damage to crops. Overall, according to the results of this experiment, it can be stated that FAS with ACCase inhibitor herbicides, such as clodinafop-propargyl, may reduce their effectiveness in controlling winter wild oat. FBS with auxin type herbicides, such as 2,4-D plus MCPA, reduce their efficacy in controlling turnipweed. FBS for clodinafop-propargyl and FAS for 2,4-D plus MCPA had a slight adverse effect on the herbicide efficacy and selectivity index. Finally, research on the use of adjuvants, alternate herbicides, and various herbicide formulations to control grass and broadleaf weeds in Iranian wheat production is suggested to prevent the adverse effects of freezing stress on herbicides.

## Author Contributions

**Conceptualization:** Mehdi Rastgoo, Bhagirath Singh Chauhan.

**Formal analysis:** Alireza Hasanfard.

**Methodology:** Alireza Hasanfard.

**Project administration:** Mehdi Rastgoo.

**Resources:** Alireza Hasanfard.

**Software:** Alireza Hasanfard.

**Supervision:** Mehdi Rastgoo, Ebrahim Izadi Darbandi, Ahmad Nezami, Bhagirath Singh Chauhan.

**Validation:** Ebrahim Izadi Darbandi, Ahmad Nezami.

**Writing – original draft:** Alireza Hasanfard.

**Writing – review & editing:** Bhagirath Singh Chauhan.

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
