## [Decision Letter · Decision Letter 0]

11 Nov 2021

PONE-D-21-33766Freezing stress affects the efficacy of clodinafop-propargyl and 2,4-D plus MCPA on wild oat and turnipweed in wheatPLOS ONE

Dear Dr. Rastgoo,

Thank you for submitting your manuscript to PLOS ONE. After careful consideration, we feel that it has merit but does not fully meet PLOS ONE’s publication criteria as it currently stands. Therefore, we invite you to submit a revised version of the manuscript that addresses the points raised during the review process.

I hope you will be able to rewrite some parts of your manuscript as defined by reviewers.

We look forward to receiving your revised manuscript.

Kind regards,

Ahmet Uludag, Ph.D.

Academic Editor

PLOS ONE

Journal Requirements:

"No"

"We are grateful to Ferdowsi University of Mashhad, Iran, for providing financial support for present research (Project No.3.49759)."

"No"

Additional Editor Comments:

Both reviewers have given very good hints to make this paper more valuable. I hope you will follow their suggestions. I think a month is enough to make all these. However, I can give you longer duration if you need. I am not going to rewrite reviewers words here.

Reviewers' comments:

Reviewer's Responses to Questions

**Comments to the Author**

1. Is the manuscript technically sound, and do the data support the conclusions?

Reviewer #1: Yes

Reviewer #2: Yes

2. Has the statistical analysis been performed appropriately and rigorously? 

Reviewer #1: Yes

Reviewer #2: Yes

3. Have the authors made all data underlying the findings in their manuscript fully available?

Reviewer #1: Yes

Reviewer #2: Yes

4. Is the manuscript presented in an intelligible fashion and written in standard English?

Reviewer #1: Yes

Reviewer #2: Yes

5. Review Comments to the Author

Reviewer #1: Unpredicted variability in temperature is associated with frequent extreme low-temperature events. And it is threatening global food security. Due to the lack of information on the efficacy of herbicides under freezing stress, this research was conducted to evaluate the efficacy of clodinafop-propargyl and 2,4-D plus MCPA on wild oat and turnipweed under freezing stress before and after herbicide application. The research results have guiding significance for production, but the article still needs to be revised and improved.

（1）The introduction part should increase the impact of global climate change on the occurrence of low temperature adversity events. And its impact on wheat production.

（2）Experimental site and plant material part：Explain the amount of fertilizer used and the ratio of N, P, K.

（3）The source of the meteorological data in Figure 1 should be clearly explained.

（4）Results part: The description of the results needs to be more in-depth. For example, how much does FBS significantly increase or decrease compared to NF?

（5）The discussion part is not deep enough, some physiological discussions should be added.

（6）The description in the conclusion part needs to be refined to increase readability.

（7）There are grammatical and editorial issues somewhere in this paper. Many of them could cause difficulties to the readers.

（8）The article does not have data on wheat yield. It would be very good if we could increase the data on the effects of herbicides before and after low temperature on wheat yield.

Reviewer #2: Dear Authors,

Reviewer comments PONE-D-21-33766

The manuscript entitled „Freezing stress affects the efficacy of clodinafop-propargyl and 2,4-D plus MCPA on wild oat and turnipweed in wheat“ represents a useful study focused on the impact of freezing stress on the efficiency of two herbicides treatments, clodinafop-propargyl and 2,4-D plus MCPA treatments, resulting in the statement that freezing stress attenuates the efficiency of herbicides in weed elimination as indicated by a decreased selectivity index. This result also indicates that freezing increases potential harmfulness of these herbicides for wheat as a crop.

I have only a few minor comments on the present manuscript:

In Materials and methods, Statistical analysis, as well as in Results, Table 1, a kind of a post-hoc test following ANOVA analysis used for determination of significant differences, has to be specified both in Materials and methods, Statistical analysis section and in results, Table 1 legend. Moreover, in Table 1 legend, it has to be clearly stated that the presented data represent mean±SE, and the number of replicates has to be specified.

Further formal comments on the manuscript text:

Abstract, lines 37-38: Add a comma both preceding and following the words „for 2,4-D plus MCPA,…“

Introduction, line 53: Use the term „winter cereals“ instead of „autumn cereals“ for autumn-sown cereals which have vernalization requirement thus they require winter temperatures prior to flowering.

Introduction, line 59: Use the term „optimum climatic conditions“ instead of „normal climatic conditions“.

Materials and methods, line 109: Replace the verb „obtain“ with „induce“ in the statement „To induce plant cold acclimation, the plants were kept outdoors,…“

Materials and methods, line 125: The authors wrote that „Freezing treatment was performed using a thermogradient freezer.“ However, basic information on the freezer has to be given including the manufacturer (company). Was the freezing treatment always just to -4 °C?

Line 175: Remove „the“ preceding the term „relative potency“.

Line 182: Mdify the statement as follows: „Relative potency < 1 indicates the decreased effectiveness of the herbicides under freezing stress with respect to optimum conditions.“

Results, Figure 4 legend, line 209; Figure 5 legend, line 246: Write rather „64 g ha-1“, not „64 g ai ha-1“.

Results, line 252: Use a plural form of the verb „were“ instead of „was“ in the statement „The GR50 of NF, FBS, and FAS were 176, 299, and 208 g ha-1, respectively…“

Line 263: Remove „the“ preceding the term „selectivity index“ and modify the statement as follows: „As shown in Table 2, selectivity index values for clodinafop-propargyl in NF, FBS, and FAS treatments were 2.4, 0.91, and 0.78, respectively, and, for 2,4-D plus MCPA, they were 2.6, 0.12, and 0.88, respectively.“

Discussion, line 299: Remove „the“ Preceding the chemical name „clodinafop propargyl“.

Discussion, line 332: Replace the word „less“ with „lower“ in the statement „The relative potency values of clodinafop-propargyl and 2,4-D plus MCPA were lower than one under both FBS and FAS treatments in the present experiment.“

Discussion, line 343: Modify the statement into a plural form as follows: „The LD10 values of 2,4-D plus MCPA in NF were 89% and 48% higher compared to FBS and FAS, respectively.“

Final recommendation: Accept after a minor revision.

6. PLOS authors have the option to publish the peer review history of their article (what does this mean?). If published, this will include your full peer review and any attached files.

Reviewer #1: No

Reviewer #2: No

---

## [Author Response · Author response to Decision Letter 0]

8 Feb 2022

Manuscript Number: PONE-D-21-33766

Manuscript title: Freezing stress affects the efficacy of clodinafop-propargyl and 2,4-D plus MCPA on wild oat and turnipweed in wheat

Dear Editor 

Thank you very much for arranging the review of our manuscript. We greatly appreciate the reviewers for their complimentary comments and suggestions. We hope that you find our responses satisfactory and that the manuscript is now acceptable.

Sincerely, 

Corresponding author

Response to the comments of the reviewer #1

We would like to thank the reviewer for the careful and thorough reading of this manuscript and the thoughtful comments and constructive suggestions, which help improve the manuscript quality. As indicated below, we have checked all the general and specific comments provided by the referees and have made necessary changes according to their indications, which are highlighted in yellow.

Reviewer #1: Unpredicted variability in temperature is associated with frequent extreme low-temperature events. And it is threatening global food security. Due to the lack of information on the efficacy of herbicides under freezing stress, this research was conducted to evaluate the efficacy of clodinafop-propargyl and 2,4-D plus MCPA on wild oat and turnipweed under freezing stress before and after herbicide application. The research results have guiding significance for production, but the article still needs to be revised and improved.

（1）The introduction part should increase the impact of global climate change on the occurrence of low temperature adversity events. And its impact on wheat production.

- The requested information has been added to the text.

（2）Experimental site and plant material part：Explain the amount of fertilizer used and the ratio of N, P, K.

- We did not use fertilizer in this experiment.

（3）The source of the meteorological data in Figure 1 should be clearly explained.

- The requested information has been added to the text.

（4）Results part: The description of the results needs to be more in-depth. For example, how much does FBS significantly increase or decrease compared to NF?

- We respect the reviewer's opinion, but according to the results of the clodinafop-propargyl, we have written the requested information in another expression (gray highlight), and its re-expression is repetitive. But in the case of 2,4-D plus MCPA, the results were explained in more detail, and the requested information was added to the text (yellow highlight).

（5）The discussion part is not deep enough, some physiological discussions should be added.

- The requested information has been added to the text.

（6）The description in the conclusion part needs to be refined to increase readability.

- The requested information was added to the text. Clear text added.

（7）There are grammatical and editorial issues somewhere in this paper. Many of them could cause difficulties to the readers.

- We respect the reviewer's opinion, but a Native English-speaker colleague has edited the manuscript's text. We rechecked the text.

（8）The article does not have data on wheat yield. It would be very good if we could increase the data on the effects of herbicides before and after low temperature on wheat yield.

- It is an exciting suggestion. We are going to study the wheat yield in future studies. In this study, we focused on the effect of freezing stress on herbicide efficacy in the early stages of plant species. In general, it is predicted that wheat yield will decrease due to herbicide efficiency and its adverse effects on the crop (selectivity index).

Response to the comments of the reviewer #2

We greatly appreciate the reviewer's positive comments. As indicated below, we have checked all the general and specific comments provided by the referees and have made necessary changes according to their indications, which are highlighted in yellow.

Reviewer #2: Dear Authors,

Reviewer comments PONE-D-21-33766

The manuscript entitled „Freezing stress affects the efficacy of clodinafop-propargyl and 2,4-D plus MCPA on wild oat and turnipweed in wheat“ represents a useful study focused on the impact of freezing stress on the efficiency of two herbicides treatments, clodinafop-propargyl and 2,4-D plus MCPA treatments, resulting in the statement that freezing stress attenuates the efficiency of herbicides in weed elimination as indicated by a decreased selectivity index. This result also indicates that freezing increases potential harmfulness of these herbicides for wheat as a crop.

I have only a few minor comments on the present manuscript:

In Materials and methods, Statistical analysis, as well as in Results, Table 1, a kind of a post-hoc test following ANOVA analysis used for determination of significant differences, has to be specified both in Materials and methods, Statistical analysis section and in results, Table 1 legend. Moreover, in Table 1 legend, it has to be clearly stated that the presented data represent mean±SE, and the number of replicates has to be specified.

- The requested information has been added to the text and Table 1.

Further formal comments on the manuscript text:

Abstract, lines 37-38: Add a comma both preceding and following the words „for 2,4-D plus MCPA,…“

- The requested revision was applied to the text.

Introduction, line 53: Use the term „winter cereals“ instead of „autumn cereals“ for autumn-sown cereals which have vernalization requirement thus they require winter temperatures prior to flowering.

- The requested revision was applied to the text.

Introduction, line 59: Use the term „optimum climatic conditions“ instead of „normal climatic conditions“.

- The requested revision was applied to the text.

Materials and methods, line 109: Replace the verb „obtain“ with „induce“ in the statement „To induce plant cold acclimation, the plants were kept outdoors,…“

- The requested revision was applied to the text.

Materials and methods, line 125: The authors wrote that „Freezing treatment was performed using a thermogradient freezer.“ However, basic information on the freezer has to be given including the manufacturer (company). Was the freezing treatment always just to -4 °C?

- The requested information was added to the text.

- Trend of temperature decrease in thermogradient freezer during the night from 7:00 P.M. to 5:00 A.M. (Please see Fig. 2 and gray highlight)

Time Temperature (℃)

7 P.M. 5

12 A.M. Reach a temperature of 0

1 A.M. kept at 0

2 A.M. Reach a temperature of -2

3 A.M. kept at -2

4 A.M. Reach a temperature of -4

5 A.M. kept at -4

5:30 A.M. Removed from the freezer

Line 175: Remove „the“ preceding the term „relative potency“.

- The requested revision was applied to the text.

Line 182: Modify the statement as follows: „Relative potency < 1 indicates the decreased effectiveness of the herbicides under freezing stress with respect to optimum conditions.“

- The requested revision was applied to the text.

Results, Figure 4 legend, line 209; Figure 5 legend, line 246: Write rather „64 g ha-1“, not „64 g ai ha-1“.

- The requested revision was applied to the text.

Results, line 252: Use a plural form of the verb „were“ instead of „was“ in the statement „The GR50 of NF, FBS, and FAS were 176, 299, and 208 g ha-1, respectively…“

- The requested revision was applied to the text.

Line 263: Remove „the“ preceding the term „selectivity index“ and modify the statement as follows: „As shown in Table 2, selectivity index values for clodinafop-propargyl in NF, FBS, and FAS treatments were 2.4, 0.91, and 0.78, respectively, and, for 2,4-D plus MCPA, they were 2.6, 0.12, and 0.88, respectively.“

- The requested revision was applied to the text.

Discussion, line 299: Remove „the“ Preceding the chemical name „clodinafop propargyl“.

- The requested revision was applied to the text.

Discussion, line 332: Replace the word „less“ with „lower“ in the statement „The relative potency values of clodinafop-propargyl and 2,4-D plus MCPA were lower than one under both FBS and FAS treatments in the present experiment.“

- The requested revision was applied to the text.

Discussion, line 343: Modify the statement into a plural form as follows: „The LD10 values of 2,4-D plus MCPA in NF were 89% and 48% higher compared to FBS and FAS, respectively.“

- The requested revision was applied to the text.

We look forward to hearing from you regarding our submission. We would be glad to respond to any further questions and comments that you may have.

---

## [Decision Letter · Decision Letter 1]

4 Apr 2022

PONE-D-21-33766R1Freezing stress affects the efficacy of clodinafop-propargyl and 2,4-D plus MCPA on wild oat and turnipweed in wheatPLOS ONE

Dear Dr. Rastgoo,

Thank you for submitting your manuscript to PLOS ONE. After careful consideration, we feel that it has merit but does not fully meet PLOS ONE’s publication criteria as it currently stands. Therefore, we invite you to submit a revised version of the manuscript that addresses the points raised during the review process. Your reasoning why you have chosen mix of two auxinic ingredients and discussion needs to be improved although it has been successful already. You need to use latin names of weeds in title and abstract as well. You need further information to be given about herbicides such as safener in clodinafop if there is, acid equvalent rates of auxicinic herbicides, separated amounts of 2,4-D and MCPA. I am not sure if you need both ED50 and GR50. By the way, no GR50 calculation in your materials and methods.  Please see comments in this e mail and attched sanitized manuscripts.  

We look forward to receiving your revised manuscript.

Kind regards,

Ahmet Uludag, Ph.D.

Academic Editor

PLOS ONE

Journal Requirements:

Additional Editor Comments (if provided):

The topic and manuscript are good in general. However, there are some more issues needs to be changed in your manuscript and issues needs to be explained further. I was not able to find out the number of weeds (density) in experimental area. This can help you explain what you thought 93% control is not acceptable if you have very high numbers of weeds. Especially suggestions of reviewer three on discussion session needs special care.

Please do not skip my personal suggestions either.

Reviewers' comments:

Reviewer's Responses to Questions

**Comments to the Author**

1. If the authors have adequately addressed your comments raised in a previous round of review and you feel that this manuscript is now acceptable for publication, you may indicate that here to bypass the “Comments to the Author” section, enter your conflict of interest statement in the “Confidential to Editor” section, and submit your "Accept" recommendation.

Reviewer #2: All comments have been addressed

Reviewer #3: (No Response)

2. Is the manuscript technically sound, and do the data support the conclusions?

Reviewer #2: Yes

Reviewer #3: Partly

3. Has the statistical analysis been performed appropriately and rigorously? 

Reviewer #2: Yes

Reviewer #3: Yes

4. Have the authors made all data underlying the findings in their manuscript fully available?

Reviewer #2: Yes

Reviewer #3: Yes

5. Is the manuscript presented in an intelligible fashion and written in standard English?

Reviewer #2: Yes

Reviewer #3: Yes

6. Review Comments to the Author

Reviewer #2: Reviewer comments PONE-D-21-33766R1

The revised manuscript entitled „Freezing stress affects the efficacy of clodinafop-propargyl and 2,4-D plus MCPA on wild oat and turnipweed in wheat“ was significantly improved by the authors according to my previous comments.

I think that the revised manuscript is suitable for publication in PLoS One.

I have only a few formal comments on the revised manuscript which are provided below:

Abstract, line 23: The abbreviations „2,4-D“ and „MCPA“ have to be explained when used for the first time.

Abstract, line 29 and the further text including the figures: The „ai“ used in the „g ai ha-1“ has to be explained. What is the difference between „g ai ha-1“ and „g ha-1“??

Line 134: Add a comma following the words „Each morning at 5:30 A.M., …“

Line 337: Use the term „optimal non-freezing conditions“ instead of „normal conditions.“

Final recommendation: Accept after a formal revision.

Reviewer #3: In general, the paper is interesting and the topic nice. However, several serious issues ought to be addressed. First of all, I absolutely disagree with statements of the authors (in the abstract and the text) that the occurrence of freezing stress around herbicides application is one of the most important factors influencing their performance. Many more factors (rate, growth stage, adjuvant, weed species, water volume, pressure etc) play a significant role on that. Moreover, survival 7% means efficacy of 93%, farmers do not need to double the rate to achieve a complete control (usually farmers are satisfied with an efficacy of 90-95%). In wild oat authors say that under no freezing (NF) and freezing after spray (FAS) conditions, winter wild oat was completely controlled with the recommended dose of clodinafop-propargyl, while at FBS control was 93%. Why the ranking of LD50 and GR50 was different (meaning the FBS between the two others)? There is an important confusion. Furthermore, I would encourage authors a) to justify their choice to evaluate an ACCase inhibitor alone and a mixture of auxinic herbicides (and not an individual)? and b) substantially revise and improve the discussion section in order to talk more about the potential mechanism behind that reduced efficacy and the potential generalization of their results (since in many cases farmers chose to spray under low temperatures since the notice an increased susceptibility of several weeds to glyphosate and other herbicides). Consequently, I recommend a major revision of the paper.

7. PLOS authors have the option to publish the peer review history of their article (what does this mean?). If published, this will include your full peer review and any attached files.

Reviewer #2: **Yes: **Klára Kosová

Reviewer #3: No

---

## [Author Response · Author response to Decision Letter 1]

27 Apr 2022

Manuscript Number: PONE-D-21-33766

Manuscript title: Freezing stress affects the efficacy of clodinafop-propargyl and 2,4-D plus MCPA on wild oat and turnipweed in wheat

Dear Editor 

We sincerely appreciate all valuable comments and suggestions, which helped us to improve the quality of the article. We hope that you find our responses satisfactory and that the manuscript is now acceptable.

Sincerely, 

Mehdi Rastgoo

Associate professor,

Department of Agrotechnology, Faculty of Agriculture, Ferdowsi University of Mashhad, Iran

Corresponding author

General comment

Your reasoning why you have chosen mix of two auxinic ingredients and discussion needs to be improved although it has been successful already. You need to use latin names of weeds in title and abstract as well. You need further information to be given about herbicides such as safener in clodinafop if there is, acid equvalent rates of auxicinic herbicides, separated amounts of 2,4-D and MCPA. I am not sure if you need both ED50 and GR50. By the way, no GR50 calculation in your materials and methods. 

- The mix of these two auxinic ingredients is one of the most important compounds in the control of broadleaf weeds in wheat fields globally, whose efficiency is generally affected by abiotic stresses. We feel that we have answered this question well in the text of the paper.

- Latin names of weeds were added as Titles and Abstracts.

- We respect the reviewer's opinion, but it is not common to write its full name in a scientific article. However their full names are as follows.

- 2,4-Dichlorophenoxyacetic acid and 2-methyl-4-chlorophenoxyacetic acid

- In this experiment, we calculated two important survival and dry weight parameters. We calculated both parameters because sometimes the herbicide does not lead to complete control of the weed plant but reduces the growth of the plant and its dry weight. Therefore, the optimal effect of the herbicide may be hidden in reducing the dry weight of the herbicide.

- In the Dose-response experiments section, we have mentioned how to calculate both parameters (LD50 and GR50).

Response to the comments of the reviewer #2

We would like to thank the reviewer for the careful and thorough reading of this manuscript and the thoughtful comments and constructive suggestions, which help improve the manuscript quality. As indicated below, we have checked all the general and specific comments provided by the referees and have made necessary changes according to their indications, which are highlighted in yellow.

Reviewer #2: Reviewer comments PONE-D-21-33766R1

The revised manuscript entitled „Freezing stress affects the efficacy of clodinafop-propargyl and 2,4-D plus MCPA on wild oat and turnipweed in wheat“ was significantly improved by the authors according to my previous comments.

I think that the revised manuscript is suitable for publication in PLoS One.

I have only a few formal comments on the revised manuscript which are provided below:

Abstract, line 23: The abbreviations „2,4-D“ and „MCPA“ have to be explained when used for the first time.

- We respect the reviewer's opinion, but this is the common name of the herbicide and should be mentioned without further explanation. It is not common to write its full name in a scientific article. However their full names are as follows.

2,4-Dichlorophenoxyacetic acid and 2-methyl-4-chlorophenoxyacetic acid

Abstract, line 29 and the further text including the figures: The „ai“ used in the „g ai ha-1“ has to be explained. What is the difference between „g ai ha-1“ and „g ha-1“??

- Both units are correct. Usually, g ai ha-1 is first mentioned in articles and then g ha-1.

Line 134: Add a comma following the words „Each morning at 5:30 A.M., …“

- The requested revision was applied to the text.

Line 337: Use the term „optimal non-freezing conditions“ instead of „normal conditions.“

- The requested revision was applied to the text.

Response to the comments of the reviewer #3

In general, the paper is interesting and the topic nice. However, several serious issues ought to be addressed. First of all, I absolutely disagree with statements of the authors (in the abstract and the text) that the occurrence of freezing stress around herbicides application is one of the most important factors influencing their performance. Many more factors (rate, growth stage, adjuvant, weed species, water volume, pressure etc) play a significant role on that. Moreover, survival 7% means efficacy of 93%, farmers do not need to double the rate to achieve a complete control (usually farmers are satisfied with an efficacy of 90-95%). In wild oat authors say that under no freezing (NF) and freezing after spray (FAS) conditions, winter wild oat was completely controlled with the recommended dose of clodinafop-propargyl, while at FBS control was 93%. Why the ranking of LD50 and GR50 was different (meaning the FBS between the two others)? There is an important confusion. Furthermore, I would encourage authors a) to justify their choice to evaluate an ACCase inhibitor alone and a mixture of auxinic herbicides (and not an individual)? and b) substantially revise and improve the discussion section in order to talk more about the potential mechanism behind that reduced efficacy and the potential generalization of their results (since in many cases farmers chose to spray under low temperatures since the notice an increased susceptibility of several weeds to glyphosate and other herbicides). 

- Many factors affect the effectiveness of herbicides. We also mentioned throughout the text that frost stress is "one of the most important factors" influencing the performance of herbicides.

- We completely agree with the esteemed referee regarding the ideal amount of weed control at the recommended dose (90 to 95%). But we did not know the result before the test, and naturally, we had to try twice the recommended dose. However, it is not recommended with 93% control of double dose use, and we discussed this issue in the article.

- In non-stress freezing and freezing after spraying with the recommended dose of clodinafop-propargil, the survival rate was zero (Figure 4), but the results were different for LD50 and GR50 (Table 1). Survival % and survival and dry weight 50% (LD50 and GR50) can vary according to their definition (more details in the results and discussion section). Survival % with LD50 and GR50 can differ depending on their definition.

- More information in the results and discussion section.

We look forward to hearing from you regarding our submission. We would be glad to respond to any further questions and comments that you may have.

---

## [Decision Letter · Decision Letter 2]

21 Jul 2022

PONE-D-21-33766R2Freezing stress affects the efficacy of clodinafop-propargyl and 2,4-D plus MCPA on wild oat (Avena ludoviciana Durieu) and turnipweed [Rapistrum rugosum (L.) All.] in wheatPLOS ONE

Dear Dr. Rastgoo,

Thank you for submitting your manuscript to PLOS ONE. After careful consideration, we feel that it has merit but does not fully meet PLOS ONE’s publication criteria as it currently stands. Therefore, we invite you to submit a revised version of the manuscript that addresses the points raised during the review process.

You will see my comments  below part of this e mail. I would like to say repeating issue is  still acceptable bu Plosone. Could you please check all suggestions and recheck any missing point from the earliest reviews. 

We look forward to receiving your revised manuscript.

Kind regards,

Ahmet Uludag, Ph.D.

Academic Editor

PLOS ONE

Additional Editor Comments (if provided):

In each cycle, a reviewer ended up with major revision. It think the reason is writing style, statistical problems, not strong interpretations. It does not mean totally unacceptable but, makes your paper problematic. I think you need some editorial improvement in your manuscript as reviewer's mentioned although this is not a problem of text throughout. Could you please check again your statistical analyses with a help a biometrician to make sure your interpretations. In addition, reviewer 4 has pointed out a few misleading statements in the manuscript, which need to be revised according to my suggestions. Furthermore, one more time visit earlier suggestions on earlier version's of the manuscript.

Reviewers' comments:

Reviewer's Responses to Questions

**Comments to the Author**

1. If the authors have adequately addressed your comments raised in a previous round of review and you feel that this manuscript is now acceptable for publication, you may indicate that here to bypass the “Comments to the Author” section, enter your conflict of interest statement in the “Confidential to Editor” section, and submit your "Accept" recommendation.

Reviewer #4: (No Response)

Reviewer #5: All comments have been addressed

2. Is the manuscript technically sound, and do the data support the conclusions?

Reviewer #4: Partly

Reviewer #5: Yes

3. Has the statistical analysis been performed appropriately and rigorously? 

Reviewer #4: No

Reviewer #5: Yes

4. Have the authors made all data underlying the findings in their manuscript fully available?

Reviewer #4: Yes

Reviewer #5: Yes

5. Is the manuscript presented in an intelligible fashion and written in standard English?

Reviewer #4: No

Reviewer #5: Yes

6. Review Comments to the Author

Reviewer #4: The manuscript investigated the response of two problematic weed species to herbicides under freezing conditions. I have realized that some other colleagues had reviewed the manuscript. I also realized that one of the reviewers raised a concern regarding the language of the manuscript, but the authors rebutted this comment and claimed that their manuscript had been edited by “a Native English-speaker colleague”. Please note that being a native English speaker is not synonymous with being a good writer. After reading the manuscript, I also feel the language is not comprehensive in many places and does not meet the minimum standard required by international journals. Hence, I recommend editing help from a language service.

Specific comments

Lines 24 and 68- Please spell out these abbreviated herbicide names. The idea is that the manuscript should be comprehensive to any readers, not only to those who share the same area of expertise as the authors.

Line 100 “seeds of winter wild oat and turnipweed” How many plants did you sample? Did the samples size sufficiently represent the population?

Line 105 “The seeds were placed in” I assume this refers to weed seed germination. How about wheat?

Line 109 “same time on October 21 (first run) and 24 (second run) in 2019”. I would not call this “a repeat in time”. We usually repeat experiments in time to cover sufficient spatial and temporal variation. Repeating an experiment 3 days after the other does not meet this criterion.

Line 111 Nutrient deficiency can affect herbicide efficiency. The authors claimed they had not added any fertilizer to the potting media. This needs to be justified in the manuscript; otherwise, it is a major flaw.

Line 112 Inconsistency. In Line 106-108, the authors claimed that they had pregerminated the seed of each weed species, and subsequently, they transplanted the seedlings. But here, they claimed they planted the seed directly. Please justify.

Line 113 Only eight plants per replicate? Would this number of plants sufficiently represent the population sample? Since the authors estimated the LD50, I suspect the low number of plants used in this experiment would sufficiently represent the variability among individuals of sampled populations? How many pregerminated seeds did you place in each pot for turnipweed?

Line 115 “irrigated according to daily requirements” How?

Line 114 How did you grow wheat?

Lines 132-133 “The pots were kept at -4℃ for 1 h and then removed from the freezer (Fig. 2).” I can`t follow this part. The plants were supposed to be kept in a freezer from 7pm to 5 am. But the authors remove the plants after 1 h exposure to -4 °C. Does this mean that the temperature reached to 4 °C at about 4 am?

Line 157 “herbicides” Please give more details than “herbicides”. What was the formulation? Did you use any wetting agents? What was the trade name of each herbicide? What was the portion of each herbicide in the mixture of 2,4-D and MCPA?

Line 174 “the curve around e, and e is the effective dose”, The letter “e” is used in two different places in the equation. Which one represents the “effective dose”?

Line 195- 196 “Afterward, differences between treatments …” This is not really a solid approach to compare treatments statistically.

Line 197-198 Pooling data from two experiments is a poor practice and results in missing many effects that are needed. The most appropriate statistical approach is to use a “mixed model analysis”. Please see the manuscript below for more information.

Piepho, H. P., A. Bu¨chse, and K. Emrich, 2003: A hitchhiker’s guide to the mixed model analysis of

randomized experiments. J. Agron. Crop Sci. 189, 310—322.

Line 207, As one of the reviewers noted, 7% is really negligible in terms of herbicide efficacy. But As noted by the authors (Table 2), it appears that freezing treatments reduced the tolerance of wheat to the ACCase inhibiting herbicide, suggesting using 2-fold the recommended rate would definitely damage the crop.

Line 228 In Table 1, the authors indicated the significance levels using asterisks; however, it is not clear what the asterisks represent. I assume they represent the significance of the parameters in the model rather than the differences between treatments. The authors need to use an appropriate statistical analysis to compare the estimated parameters between treatments, and the significant differences between treatments can be demonstrated using letters “e.g. a, b, etc.).

Line 272 “However, LD10 of wheat in” The authors must include the dose-response results for wheat treated with both herbicides under all three conditions.

Line 286 “nights before and after spray reduces the efficacy of this herbicide compared to NF.” This statement is misleading. First of all, the differences in LD50 values appear to be negligible. Secondly, there is natural variation among the individuals of a population resulting in varied responses to herbicides. So such a low difference can be due to variability among individuals.

Lines 289-289 “Freezing treatment for three nights before and after spraying of clethodim herbicide (the mode of action similar to clodinafop-propargyl) led to an increase in LD50 “. In the manuscript by Saini et al. 2017, the authors claimed that “Addition of frost treatments did not markedly affect survival of the S population” and eventually noted no significant differences. I feel in the current manuscript, the authors have misinterpreted their results.

Line 300- 301 “Herbicide translocation in the phloem in freezing conditions could decline due to damage to the phloem cells” The freezing treatment only lasted for 3 days; thus, I suspect impaired translocation could explain the outcome of this research. I guess in your case, freezing temperature reduced herbicide absorption.

Lines 314-322, 333-337 and 360-361 The physiological mechanisms underlying the response of plants to frost are different from those of low (cold) temperatures. So the response of plants to herbicides would be different under freezing versus low temperatures. For instance, for turnipweed, the freezing treatment before herbicide application reduced the efficacy of 2,4-d and MCPA treatment later. As there were no herbicide molecules in the plants at the time of freezing treatment, reduced translocation of the herbicide can not solely explain reduced herbicide efficacy in FBS plants.

Line 370 “their effectiveness in controlling narrow leaf…” you only studied one grass species. So this may not be true for other “narrow-leaved” species.

Line 374-375 “Therefore, the use of these herbicides in the mentioned conditions (FBS for clodinafop-propargyl and FAS for 2,4-D plus MCPA) is recommended.” This is misleading, as it appears the desirable crop (wheat) loses its tolerance to both herbicides under both freezing conditions. So, it is not a good idea to use these herbicides under freezing conditions to control weed species in wheat. Please revise accordingly.

Reviewer #5: The authors incorporated the comments of both reviewers in true sense. The revised draft shaped well and can be accepted for publication.

7. PLOS authors have the option to publish the peer review history of their article (what does this mean?). If published, this will include your full peer review and any attached files.

Reviewer #4: No

Reviewer #5: No

---

## [Author Response · Author response to Decision Letter 2]

25 Jul 2022

Manuscript Number: PONE-D-21-33766

Manuscript title: Freezing stress affects the efficacy of clodinafop-propargyl and 2,4-D plus MCPA on wild oat (Avena ludoviciana Durieu) and turnipweed [Rapistrum rugosum (L.) All.] in wheat (Triticum aestivum L.)

Dear Dr Uludag

Thank you very much for organizing the review of our manuscript. We have addressed all the comments/issues raised by the reviewers and have made necessary changes according to their indications, which are highlighted in purple (new revision).

The opinions of reviewer 4 are extraordinary, and we do not agree with some of them. Our manuscript has been read several times by a Native English speaker and recognized weed science expert. He emphasized that there was no problem with the language of this manuscript. The reviewer should not criticize the language of the article with a general comment. If the language has a problem, it should mention exactly which sentence? Which section?

Our MS has been read several times by one of the top experts and editors in weed science (professor Chauhan). Isn't this the reviewer's expression strange?!

Reviewer 4: Note that being a native English speaker is not synonymous with being a good writer.

His opinion about statistical analysis shocked us. We used the most powerful method and software for analysis. Regarding the data pooling, the variance of the data of the two experiments is equal (based on Bartlett's or Levene’s test), and we can pool the data. You will find a simple search for high-quality sources. Data for both runs were pooled as no significant differences were observed between runs.

- Arsham, H., & Lovric, M. (2011). Bartlett's Test. International encyclopedia of statistical science, 1, 87-88.

- Niyonzima, N., Bakke, S. S., Gregersen, I., Holm, S., Sandanger, Ø., Orrem, H. L., ... & Espevik, T. (2020). Cholesterol crystals use complement to increase NLRP3 signaling pathways in coronary and carotid atherosclerosis. EBioMedicine, 60, 102985.

-Mahajan, G., Rachaputi, R. C. N., & Chauhan, B. S. (2020). Horse purslane (Trianthema portulacastrum) control in pigeonpea with PRE and POST herbicides. Weed Technology, 34(5), 764-769.

-Mobli, A., Manalil, S., Khan, A. M., Jha, P., & Chauhan, B. S. (2020). Effect of emergence time on growth and fecundity of Rapistrum rugosum and Brassica tournefortii in the northern region of Australia. Scientific Reports, 10(1), 1-10.

- Sritan, K., & Phuenaree, B. (2021). A Comparison of Efficiency for Homogeneity of Variance Tests under Log-normal Distribution. Asian Journal of Applied Sciences, 9(4).

We should not rely on a particular article from 19 years ago. We have explained this in a clear and complete analysis section. Note that none of the reviewers objected to our analysis except reviewer 4.

In all high-quality research, the use of standard error mean (SEM) is expected. 

Saini, R. K., Malone, J., Preston, C., & Gill, G. S. (2016). Frost reduces clethodim efficacy in clethodim-resistant rigid ryegrass (Lolium rigidum) populations. Weed Science, 64(2), 207-215.

The SEM is used to determine the differences between more than one sample of data. It helps us estimate how well your sample data represents the whole population by measuring the accuracy with which the sample data represents a population using standard deviation. However, we added items like mean comparison (LSD test) to the manuscript (Table 1).

Apart from the above, we feel that the reviewer did not read the paper well. For example, we had three repetitions in each run, and each repetition had eight plants. The reviewer incorrectly stated eight plants in each repetition. In addition, despite our complete explanations and figures, the reviewer did not notice that the plants were exposed to freezing stress for three nights, which reached the minimum temperature (-4°C).

Reviewer 4 has requested the details of herbicides. Meanwhile, we have already provided all the valuable information in the paper (purple highlights in the Introduction and Experimental factors).

Sincerely, 

Corresponding author

p.s. we have addressed the rest, and we expect to hear your final opinion after about a year and apply the comments of five reviewers. Naturally, you send each manuscript to each reviewer, who will have their unique comments and revisions. How long should this cycle continue? We emphasize that we applied all the comments of previous reviewers (as declared by reviewer 5). We hope that you find our responses satisfactory and that the manuscript is now acceptable.

Review Comments to the Author

Reviewer #4: The manuscript investigated the response of two problematic weed species to herbicides under freezing conditions. I have realized that some other colleagues had reviewed the manuscript. I also realized that one of the reviewers raised a concern regarding the language of the manuscript, but the authors rebutted this comment and claimed that their manuscript had been edited by “a Native English-speaker colleague”. Please note that being a native English speaker is not synonymous with being a good writer. After reading the manuscript, I also feel the language is not comprehensive in many places and does not meet the minimum standard required by international journals. Hence, I recommend editing help from a language service.

We expressed our opinion about this to the academic editor.

Specific comments

Lines 24 and 68- Please spell out these abbreviated herbicide names. The idea is that the manuscript should be comprehensive to any readers, not only to those who share the same area of expertise as the authors.

The requested information was added to the text (2,4-Dichlorophenoxyacetic acid and 2-methyl-4-chlorophenoxyacetic acid). The clodinafop-propargyl is the full name of this herbicide and not an abbreviation.

Line 100 “seeds of winter wild oat and turnipweed” How many plants did you sample? Did the samples size sufficiently represent the population?

Weed seeds are collected from at least 500 plants, and the sample size sufficiently represents the population.

Line 105 “The seeds were placed in” I assume this refers to weed seed germination. How about wheat?

The requested information was added to the text.

Line 109 “same time on October 21 (first run) and 24 (second run) in 2019”. I would not call this “a repeat in time”. We usually repeat experiments in time to cover sufficient spatial and temporal variation. Repeating an experiment 3 days after the other does not meet this criterion.

We did not claim that the experiment was repeated in time. The examination is designed based on our purposes.

Line 111 Nutrient deficiency can affect herbicide efficiency. The authors claimed they had not added any fertilizer to the potting media. This needs to be justified in the manuscript; otherwise, it is a major flaw.

We agree with the reviewer, but a large part of the soil used was farm soil, which had suitable physiochemical. Also, note that the soil conditions were the same in all experiments. Hence, the results are reported under similar conditions.

Line 112 Inconsistency. In Line 106-108, the authors claimed that they had pregerminated the seed of each weed species, and subsequently, they transplanted the seedlings. But here, they claimed they planted the seed directly. Please justify.

Both weed species were pregerminated. Then Germinated turnipweed seeds were planted in a seedling tray. Germinated wild oat seeds were planted directly in the soil. Fully and clearly explained (Please see M&M).

Line 113 Only eight plants per replicate? Would this number of plants sufficiently represent the population sample? Since the authors estimated the LD50, I suspect the low number of plants used in this experiment would sufficiently represent the variability among individuals of sampled populations? How many pregerminated seeds did you place in each pot for turnipweed?

8 plants×3Rep = (24 plants) × 2 Run= 48 plants

It makes sense for a controlled trial.

Eight turnipweed seedlings transferred to pots (added to the text).

Line 115 “irrigated according to daily requirements” How?

Irrigation was sufficient (based on the soil requirement of a pot). ~ 400 ml per pot. In this way, the plants were not subjected to drought stress. 

Line 114 How did you grow wheat?

It is obvious. The wheat seed does not require to break dormancy. It does not need pre-germination either. So it was direct cultivation.

Lines 132-133 “The pots were kept at -4℃ for 1 h and then removed from the freezer (Fig. 2).” I can`t follow this part. The plants were supposed to be kept in a freezer from 7pm to 5 am. But the authors remove the plants after 1 h exposure to -4 °C. Does this mean that the temperature reached to 4 °C at about 4 am?

We have provided clear explanations in the text. We have included figures 2 and 3 in the MS for clarification. The plants were exposed to the determined temperature gradient for three nights and reached the minimum temperature (-4°C).

Trend of temperature decrease in thermogradient freezer during the night from 7:00 P.M. to 5:00 A.M. (Please see Fig. 2)

Time Temperature (℃)

7 P.M. 5

12 A.M. Reach a temperature of 0

1 A.M. kept at 0

2 A.M. Reach a temperature of -2

3 A.M. kept at -2

4 A.M. Reach a temperature of -4

5 A.M. kept at -4

5:30 A.M. Removed from the freezer

Line 157 “herbicides” Please give more details than “herbicides”. What was the formulation? Did you use any wetting agents? What was the trade name of each herbicide? What was the portion of each herbicide in the mixture of 2,4-D and MCPA?

For details on herbicides, refer to the Introduction and Experimental factors We would have mentioned it in the paper if we had used agents. So we did not use any agent. The trade name of each herbicide is added to the text (Experimental factors). . The manufacturing company combines this herbicide (360 g ae.L-1 for 2,4-D and 315 g ae.L-1 for MCPA). We did not mix it.

Line 174 “the curve around e, and e is the effective dose”, The letter “e” is used in two different places in the equation. Which one represents the “effective dose”?

Both indicate the effective dose.

Line 195- 196 “Afterward, differences between treatments …” This is not really a solid approach to compare treatments statistically.

In all high-quality research, a standard error mean (SEM) is expected. We need SEM values for a dose-response graph, and this is necessary. However, we added items like mean comparison (LSD test) to the manuscript (Table 1).

Line 197-198 Pooling data from two experiments is a poor practice and results in missing many effects that are needed. The most appropriate statistical approach is to use a “mixed model analysis”. Please see the manuscript below for more information.

Piepho, H. P., A. Bu¨chse, and K. Emrich, 2003: A hitchhiker’s guide to the mixed model analysis of

randomized experiments. J. Agron. Crop Sci. 189, 310—322.

We consulted a biometrics expert. Data pooling is correct for our research. We provided additional explanations and sources for the editor.

Line 207, As one of the reviewers noted, 7% is really negligible in terms of herbicide efficacy. But As noted by the authors (Table 2), it appears that freezing treatments reduced the tolerance of wheat to the ACCase inhibiting herbicide, suggesting using 2-fold the recommended rate would definitely damage the crop.

We agree with the reviewer. At the same time, we do not mention the use of 2-fold the recommended dose. In the Discussion section, we have discussed the essence of the topic.

Line 228 In Table 1, the authors indicated the significance levels using asterisks; however, it is not clear what the asterisks represent. I assume they represent the significance of the parameters in the model rather than the differences between treatments. The authors need to use an appropriate statistical analysis to compare the estimated parameters between treatments, and the significant differences between treatments can be demonstrated using letters “e.g. a, b, etc.).

The requested information was added to the Table 1.

Line 272 “However, LD10 of wheat in” The authors must include the dose-response results for wheat treated with both herbicides under all three conditions.

Before submitting the MS, we had drawn a dose-response diagram for wheat. It is not correct at all. In this case, we will have a graph with an almost straight line (no slope) because the reduction of survival and dry weight in wheat is much less than that of weeds. Instead, we have used LD10 and selected the index.

Line 286 “nights before and after spray reduces the efficacy of this herbicide compared to NF.” This statement is misleading. First of all, the differences in LD50 values appear to be negligible. Secondly, there is natural variation among the individuals of a population resulting in varied responses to herbicides. So such a low difference can be due to variability among individuals.

The difference between LD50 (NF and FBS) in clodinafop-propargyl is slight, and we mentioned this in the manuscript. Most of our emphasis in clodinafop-propargyl is on LD50 more in FAS. The requested information was added to the text.

Lines 289-289 “Freezing treatment for three nights before and after spraying of clethodim herbicide (the mode of action similar to clodinafop-propargyl) led to an increase in LD50 “. In the manuscript by Saini et al. 2017, the authors claimed that “Addition of frost treatments did not markedly affect survival of the S population” and eventually noted no significant differences. I feel in the current manuscript, the authors have misinterpreted their results.

See Table 2 of Saini's paper (R population). Our interpretation is correct.

Line 300- 301 “Herbicide translocation in the phloem in freezing conditions could decline due to damage to the phloem cells” The freezing treatment only lasted for 3 days; thus, I suspect impaired translocation could explain the outcome of this research. I guess in your case, freezing temperature reduced herbicide absorption.

We suspect that absorption disturbance is the main cause of efficiency reduction for the FBS. However, the reviewer's opinion is also mentioned in the discussion section.

Lines 314-322, 333-337 and 360-361 The physiological mechanisms underlying the response of plants to frost are different from those of low (cold) temperatures. So the response of plants to herbicides would be different under freezing versus low temperatures. For instance, for turnipweed, the freezing treatment before herbicide application reduced the efficacy of 2,4-d and MCPA treatment later. As there were no herbicide molecules in the plants at the time of freezing treatment, reduced translocation of the herbicide can not solely explain reduced herbicide efficacy in FBS plants.

We have made crucial assumptions, and this is justified. Differences in the response of a plant species are also possible. Hence, our justification may also be entirely correct.

Line 370 “their effectiveness in controlling narrow leaf…” you only studied one grass species. So this may not be true for other “narrow-leaved” species.

We agree with the reviewer, but please note our sentence:

Overall, according to the results of this experiment, it can be stated that FAS with ACCase inhibitor herbicides, such as clodinafop-propargyl, may reduce their effectiveness in controlling narrow leaf weeds such as winter wild oat. 

Our personal experience has shown this essentially (anecdotal evidence). However, we need more studies in this field.

Line 374-375 “Therefore, the use of these herbicides in the mentioned conditions (FBS for clodinafop-propargyl and FAS for 2,4-D plus MCPA) is recommended.” This is misleading, as it appears the desirable crop (wheat) loses its tolerance to both herbicides under both freezing conditions. So, it is not a good idea to use these herbicides under freezing conditions to control weed species in wheat. Please revise accordingly.

Among the tried conditions, we have mentioned the best treatments. Please note that for us, the Selectivity index is more critical than LD10. According to Table 2, this suggestion is better compared to other conditions. However, the reviewer's statement is also considered in different parts of the Discussion.

Reviewer #5: The authors incorporated the comments of both reviewers in true sense. The revised draft shaped well and can be accepted for publication.

We look forward to hearing from you regarding our submission. We would be glad to respond to any further questions and comments that you may have.

---

## [Decision Letter · Decision Letter 3]

15 Aug 2022

PONE-D-21-33766R3Freezing stress affects the efficacy of clodinafop-propargyl and 2,4-D plus MCPA on wild oat (Avena ludoviciana Durieu) and turnipweed [Rapistrum rugosum (L.) All.] in wheatPLOS ONE

Dear Dr. Rastgoo,

Thank you for submitting your manuscript to PLOS ONE. After careful consideration, we feel that it has merit but does not fully meet PLOS ONE’s publication criteria as it currently stands. Therefore, we invite you to submit a revised version of the manuscript that addresses the points raised during the review process.

I have read your manuscript carefully. I have seen some points confusing. I hope you can explain them or make necessary changes. Your manuscript is almost ready for acceptance.

We look forward to receiving your revised manuscript.

Kind regards,

Ahmet Uludag, Ph.D.

Academic Editor

PLOS ONE

Journal Requirements:

Additional Editor Comments:

I have shown my questions and concerns on the manuscript named editor which is attached. You can avoid concerns of reviewer 5 about tables and figures that are checked by I am. By the way you have criticized me on different reviewers. Unfortunately the same reviewers did not reviewed again. This is the biggest problem of academic editors which colleagues like publishing but not reviewing. An academic editor does not gain anything from reviewing except happiness from contributing colleagues. If I can get responds to my concerns on the text clearly and you can make necessary changes I will accept your paper immediately.

Reviewers' comments:

Reviewer's Responses to Questions

**Comments to the Author**

1. If the authors have adequately addressed your comments raised in a previous round of review and you feel that this manuscript is now acceptable for publication, you may indicate that here to bypass the “Comments to the Author” section, enter your conflict of interest statement in the “Confidential to Editor” section, and submit your "Accept" recommendation.

Reviewer #2: All comments have been addressed

Reviewer #6: (No Response)

2. Is the manuscript technically sound, and do the data support the conclusions?

Reviewer #2: Yes

Reviewer #6: Partly

3. Has the statistical analysis been performed appropriately and rigorously? 

Reviewer #2: Yes

Reviewer #6: Yes

4. Have the authors made all data underlying the findings in their manuscript fully available?

Reviewer #2: Yes

Reviewer #6: Yes

5. Is the manuscript presented in an intelligible fashion and written in standard English?

Reviewer #2: Yes

Reviewer #6: No

6. Review Comments to the Author

Reviewer #2: Reviewer comments PONE-D-21-33766R3

The revised manuscript entitled „Freezing stress affects the efficacy of clodinafop-propargyl and 2,4-D plus MCPA on wild oat (Avena ludoviciana Durieu) and turnipweed (Rapistrum rugosum (L.) All.) in wheat“ was improved by the authors in accordnace with my and other reviewers´ comments.

I have no further major comment on the revised manuscript.

I have only a few formal comments on the revised manuscript which are given below:

Introduction, line 64: Add the word „fact“ in the statement „Based on this fact,…“

Materials and methods, line 116: Add the verb „were“ in the statement „…were transferred to pots…“

Discussion, line 310: Add the word „respectively“ at the end of the statemnt „the efficacy of 2,4-D plus MCPA under FBS and FAS was 1.6 and 1.1 times lower compared to NF, respectively.“

Final recommendation: Accept.

Reviewer #6: Line 160-161: This sentence is very confusing. How did you measure the survivability % by placing the samples in an oven?

Table 1: This table is very confusing. The table mentioned “the dose of clodinafop-propargyl and 2,4-D plus MCPA required for 50% mortality (LD50) and the dose that caused 50% inhibition of growth noted by biomass (GR50) of winter wild oat and turnipweed under different freezing treatments. But where is the data for the two weeds in the table?

Table 2: Confusing. The table mentioned “sensitivity of plant species of wheat, winter wild oat, and turnipweed to clodinafop-propargyl and 2,4-D plus MCPA under different freezing treatments”. But the table presents wheat and weed data for the freezing treatment.

Figure 4: Survival (A) and dry weight (B) of winter wild oat treated with various doses of clodinafop-propargyl under different freezing treatments but why these data were not presented for the turnipweed?

Figure 5: Survival (A) and dry weight (B) of turnipweed treated with various doses of 2,4-D plus MCPA under different freezing treatments but why these data were not presented for the weed wild oat?

It is very difficult to understand the message from the current conclusion. Please reduce the length of the conclusion and focus on the most important finding of the research and take-home message.

7. PLOS authors have the option to publish the peer review history of their article (what does this mean?). If published, this will include your full peer review and any attached files.

Reviewer #2: **Yes: **Klára Kosová, Ph.D.

Reviewer #6: No

---

## [Author Response · Author response to Decision Letter 3]

16 Aug 2022

Manuscript Number: PONE-D-21-33766

Manuscript title: Freezing stress affects the efficacy of clodinafop-propargyl and 2,4-D plus MCPA on wild oat (Avena ludoviciana Durieu) and turnipweed [Rapistrum rugosum (L.) All.] in wheat (Triticum aestivum L.)

Dear Dr Uludag,

I want you to know how much I value your comments. Thank you very much for organizing the review of our manuscript. We have addressed all the comments/issues raised by the reviewers. The suggestions have improved the manuscript significantly. Our specific responses are given below. We hope to hear from you soon.

Sincerely, 

Corresponding author

I carefully reviewed your revisions/comments. I applied most of them and presented answers for some below.

Editor's comments

Is it not controversial with line 30-31? "FBS caused 7% survival" you have mentioned

- " The requested revision was added to the text. According to the results of LD50"

Is this not contraversial with abstract? As far as I understood NF and FBS caused 50% control similar for NF and FBS but FAS required more herbicide to kill 50% of the population. 

According to Table 1, You should mention in the abstract clerary the difference between two herbicides/weeds because one needs hghre rate in FAS the other in FBS.

- The last sentence of the abstract was revised based on LD50.

You can mention in the abstract that FR50 and LD50 gave the similar outcomes. I think the final interpretation will not change for both outputs.

- We mentioned this result in another way in the Abstract.

 “The LD50 (dose required to control 50% of individuals in the population) and GR50 (dose causing 50% growth reduction of plants) rankings were NF<FBS<FAS for clodinafop-propargyl and NF<FAS<FBS for 2,4-D plus MCPA.”

Your data does not says twice. It is a value between x and 2x according to tables and figures you have given. 

In addition is it FAS or FBS. I confused much.

Please see Figure 4 (A). For complete control of wild oat in FBS treatment, double the recommended dose (128 g ai ha-1) is needed, and the sentence is correct.Is it not controversial with former sentence?

- Please see Figure 4

. This sentence is also controversial with the first sentence. I think the first sentence is not correct.

- Please note that the first sentence is related to Figure 4. The discussion of LD50 is different and clarified in Table 1.

It is confusing for me FAS requires this not FBS. Am I wrong in interpretation of figure 4 and Table 1?

- Please see Figure 4 (B). The sentence is correct.

- 

This is what I have seen in your results but I mentioned above that some of your statements are controversial with this statetment. Check your abstract as well.

Check the previous sentence. According to GR50, this sentence is mentioned. The sentence was improved.When you discusss effect of herbicides under freezing use all data together. You can add this paragraph above paragraphs. Because LD50, GR50, potency, all shows the same conclusion 

I think using less word to explain your data makse it easier for the readers.

- Yes, I accept your opinion. We did this in the Conclusions. It confirms the relative potency of the previous sections.

This is eactly what your data shows. Some sentemces I have shown above are contoversial. Could you please check them carefully.

- I appreciate your attention. I carefully checked different parts of the MS. Ambiguities are resolved when we refer to figures and tables. I have made some changes to the paper.

I do not think so this is result of your experiment for clodinafop. FBS and NF is not statistically different for clodinaforp. please follow the what statstics tell.

- The sentence was deleted.

I recommend you that please avoid generilazation. You can say an ACCase inhibitor clodinafop... It helps your aim without generilazation. As you know not all chemistries have similar effects.

- Yes, your opinion is correct. Our sentence is also such that it does not refer to all the herbicides of this family.

As I mentioned above use winter wild oat, a narrow leaf weed. It helps you avoiding unnecessary genetralization.

- The requested revision was applied to the text.

I do not think you need this psrt.

- The sentence was deleted.

I think this is not discussed in your manuscript. Instead, you can say that these things should be researched. 

- The requested revision was applied to the text.

---

## [Editor Report · Decision Letter 4]

24 Aug 2022

PONE-D-21-33766R4Freezing stress affects the efficacy of clodinafop-propargyl and 2,4-D plus MCPA on wild oat (Avena ludoviciana Durieu) and turnipweed [Rapistrum rugosum (L.) All.] in wheat (Triticum aestivum L.)PLOS ONE

Dear Dr. Rastgoo,

Thank you for submitting your manuscript to PLOS ONE. After careful consideration, we feel that it has merit but does not fully meet PLOS ONE’s publication criteria as it currently stands. Therefore, we invite you to submit a revised version of the manuscript that addresses the points raised during the review process.

You will see my concerns on the attached document. SOme of them can be skipped but the others should be changed. You can differentiate. The most important point should be added is the role of safener on efficacy of clodinafop. I recommend you check old papers related to former safener of fenoxaprop.

We look forward to receiving your revised manuscript.

Kind regards,

Ahmet Uludag, Ph.D.

Academic Editor

PLOS ONE

Journal Requirements:

Additional Editor Comments:

Dear colleague, I have checked your manuscript myself because most of the issues about former version were mine. I read again your manuscript and showed my suggestions. My aim is to make this invaluable data more attractive. However, the role of safener for clodinafop efficacy and effect was skipped in the manuscript. It is definitely needed. I am not suggesting an experiment on this stage but you should discuss this and add all relevant places of the manuscript. I also wonder if there is a role of formulation especially for hormon-like herbicide. I have no idea about it, it will be nice to add such discussion points to paper if possible.
---

## [Author Response · Author response to Decision Letter 4]

27 Aug 2022

Dear Dr Uludag,

cc Dear Editor in Chief

We appreciate your efforts. As you know, we submitted the manuscript a year ago. You and six reviewers carefully reviewed the manuscript. We have applied all your comments and the reviewers. In the last version, you said that you would immediately accept our manuscript if we completed your revisions, but again you requested other revisions. This process is not normal. We re-checked the manuscript carefully and included the safeners in the manuscript.

We hope to hear from you soon.

Sincerely,

---

## [Decision Letter · Decision Letter 5]

8 Sep 2022

Freezing stress affects the efficacy of clodinafop-propargyl and 2,4-D plus MCPA on wild oat (Avena ludoviciana Durieu) and turnipweed [Rapistrum rugosum (L.) All.] in wheat (Triticum aestivum L.)

PONE-D-21-33766R5

Dear Dr. Rastgoo,

Thank you for submitting your revised manuscript to PLOS ONE; I sincerely apologise for the unusually delayed review timeframe. We’re pleased to inform you that your manuscript has been judged scientifically suitable for publication and will be formally accepted for publication once it meets all outstanding technical requirements.

Kind regards,

Emily Chenette

Editor in Chief

PLOS ONE

Additional Editor Comments (optional):

Reviewers' comments:

Reviewer's Responses to Questions

**Comments to the Author**

1. If the authors have adequately addressed your comments raised in a previous round of review and you feel that this manuscript is now acceptable for publication, you may indicate that here to bypass the “Comments to the Author” section, enter your conflict of interest statement in the “Confidential to Editor” section, and submit your "Accept" recommendation.

Reviewer #2: All comments have been addressed

2. Is the manuscript technically sound, and do the data support the conclusions?

Reviewer #2: Yes

3. Has the statistical analysis been performed appropriately and rigorously? 

Reviewer #2: Yes

4. Have the authors made all data underlying the findings in their manuscript fully available?

Reviewer #2: Yes

5. Is the manuscript presented in an intelligible fashion and written in standard English?

Reviewer #2: Yes

6. Review Comments to the Author

Reviewer #2: I have no further comments on the revised manuscript. I can now recommend its prompt publication in PLoS One.

7. PLOS authors have the option to publish the peer review history of their article (what does this mean?). If published, this will include your full peer review and any attached files.

Reviewer #2: **Yes: **Klára Kosová, Ph.D.

---

## [Editor Report · Acceptance letter]

27 Sep 2022

PONE-D-21-33766R5 

Freezing stress affects the efficacy of clodinafop-propargyl and 2,4-D plus MCPA on wild oat (*Avena ludoviciana* Durieu) and turnipweed [*Rapistrum rugosum* (L.) All.] in wheat (*Triticum aestivum* L.) 

Dear Dr. Rastgoo:

I'm pleased to inform you that your manuscript has been deemed suitable for publication in PLOS ONE. Congratulations! Your manuscript is now with our production department. 

Kind regards, 

on behalf of

Dr Emily Chenette 

Staff Editor

PLOS ONE